# DIVISION: MEMORY EFFICIENT TRAINING VIA DUAL ACTIVATION PRECISION

## ABSTRACT

Activation compressed training (ACT) has been shown to be a promising way to reduce the memory cost of training deep neural networks (DNNs). However, existing work of ACT relies on searching for optimal bit-width during DNN training to reduce the quantization noise, which makes the procedure complicated and less transparent. To this end, we propose a simple and effective method to compress DNN training. Our method is motivated by an instructive observation: *DNN backward propagation mainly utilizes the low-frequency component (LFC) of the activation maps, while the majority of memory is for caching the high-frequency component (HFC) during the training*. This indicates the HFC of activation maps is highly redundant and compressible during DNN training, which inspires our proposed Dual ActIVation PrecISION (DIVISION). During the training, DIVISION preserves the high-precision copy of LFC and compresses the HFC into a light-weight copy with low numerical precision. This can significantly reduce the memory cost without negatively affecting the precision of backward propagation such that DIVISION maintains competitive model accuracy. Experimental results show DIVISION achieves over $10\times$ compression of activation maps, and significantly higher training throughput than state-of-the-art ACT methods, without loss of model accuracy. The code is available at `https://anonymous.4open.science/r/division-5CC0/`.

## 1 INTRODUCTION

Deep neural networks (DNNs) have been widely applied to real-world tasks such as language understanding (Devlin et al., 2018), machine translation (Vaswani et al., 2017), visual detection and tracking (Redmon et al., 2016). With increasingly larger and deeper architectures, DNNs achieve remarkable improvement in representation learning and generalization capacity (Krizhevsky et al., 2012). Generally, training a larger model requires more memory resources to cache the activation values of all intermediate layers during the back-propagation[1]. For example, training a DenseNet-121 (Huang et al., 2017) on the ImageNet dataset (Deng et al., 2009) requires to cache over 1.3 billion float activation values (4.8GB) during back-propagation; and training a ResNet-50 (He et al., 2016) requires to cache over 4.6 billion float activation values (17GB). Some techniques have been developed to reduce the training cache of DNNs, such as checkpointing (Chen et al., 2016; Gruslys et al., 2016), mix precision training (Vanholder, 2016), low bit-width training (Lin et al., 2017; Chen et al., 2020) and activation compressed training (Georgiadis, 2019; Liu et al., 2022). Among these, the activation compressed training (ACT) has emerged as a promising method due to its significant reduction of training memory and the competitive learning performance (Liu et al., 2021b).

Existing work of ACT relies on quantizing the activation maps to reduce the memory consumption of DNN training, such as BLPA (Chakrabarti & Moseley, 2019), TinyScript (Fu et al., 2020) and ActNN (Chen et al., 2021). Although ACT could significantly reduce the training memory cost, the quantization process introduces noises in backward propagation, which makes the training suffer from undesirable degradation of accuracy (Fu et al., 2020). Due to this reason, BLPA requires 4-bit ACT to ensure the convergence to optimal solution on the ImageNet dataset, which has only $6\times$ compression rate[2] of activation maps (Chakrabarti & Moseley, 2019). Other works propose to

---

[1] The activation map of each layer is required for estimating the gradient during backward propagation.

[2] $6\times$ compression rate indicates the memory of cached activation maps is $1/6$ of that of normal training.

search for optimal bit-width to match different samples during training, such as ActNN (Chen et al., 2021) and AC-GC (Evans & Aamodt, 2021). Although it can moderately reduce the quantization noise and achieves optimal solution under 2-bit ACT (nearly $10\times$ compression rate), the following issues cannot be ignored. First, it is time-consuming to search for optimal bit-width during training. Second, the framework of bit-width searching is complicated and non-transparent, which brings new challenges to follow-up studies on the ACT and its real-world applications.

In this work, we propose a simple and transparent method to reduce the memory cost of DNN training. Our method is motivated by an instructive observation: *DNN backward propagation mainly utilizes the low-frequency component (LFC) of the activation maps, while the majority of memory is for the storage of high-frequency component (HFC) during the training.* This indicates the HFC of activation map is highly redundant and compressible during the training. Following this direction, we propose Dual Activation Precision (DIVISION), which preserves the high-precision copy of LFC and compresses the HFC into a light-weight copy with low numerical precision during the training. In this way, DIVISION can significantly reduce the memory cost. Meanwhile, it will not negatively affect the quality of backward propagation and could maintain competitive model accuracy.

Compared with existing work that integrates searching into learning (Liu et al., 2022), DIVISION has a more simplified compressor and decompressor, speeding up the procedure of ACT. More importantly, it reveals the compressible (HFC) and non-compressible factors (LFC) during DNN training, improving the transparency of ACT. Experiments are conducted to evaluate DIVISION in terms of memory cost, model accuracy, and training throughput. An overall comparison is given in Figure 1 (a). Our proposed DIVISION consistently outperforms state-of-the-art baseline methods in the above three aspects. The contributions of this work are summarized as follows:

- We experimentally demonstrate and theoretically prove that DNN backward propagation mainly utilizes the LFC of the activation maps. The HFC is highly redundant and compressible.
- We propose a simple framework DIVISION to effectively reduce the memory cost of DNN training via removing the redundancy in the HFC of activation maps during the training.
- Experiments on three benchmark datasets demonstrate the effectiveness of DIVISION in terms of memory cost, model accuracy, and training throughput.

## 2 PRELIMINARY

### 2.1 NOTATIONS

Without loss of generality, we consider an $L$-layer deep neural network in this work. During the forward pass, for each layer $l$ ($1 \leq l \leq L$), the activation map is calculated by

$$\mathbf{H}_l = \text{forward}(\mathbf{H}_{l-1}; \mathbf{W}_l), \tag{1}$$

where $\mathbf{H}_l$ denotes the activation map of layer $l$; $\mathbf{H}_0$ takes a mini-batch of input images; $\mathbf{W}_l$ denotes the weight of layer $l$; and $\text{forward}(\cdot)$ denotes the feed-forward operation. During the backward pass, the gradients of the loss value towards the activation maps and weights are be estimated by

$$\left[\hat{\nabla}_{\mathbf{H}_{l-1}}, \hat{\nabla}_{\mathbf{W}_l}\right] = \text{backward}(\hat{\nabla}_{\mathbf{H}_l}, \mathbf{H}_{l-1}, \mathbf{W}_l), \tag{2}$$

where $\hat{\nabla}_{\mathbf{H}_{l-1}}$ and $\hat{\nabla}_{\mathbf{H}_l}$ denote the gradient towards the activation map of layer $l-1$ and $l$, respectively; $\hat{\nabla}_{\mathbf{W}_l}$ denotes the gradient towards the weight of layer $l$; and $\text{backward}(\cdot)^3$ denotes the backward function which takes $\hat{\nabla}_{\mathbf{H}_l}$, $\mathbf{H}_{l-1}$ and $\mathbf{W}_l$, and outputs the gradients $\hat{\nabla}_{\mathbf{H}_{l-1}}$ and $\hat{\nabla}_{\mathbf{W}_l}$. Equation (2) indicates it is required to cache the activation maps $\mathbf{H}_0, \cdots, \mathbf{H}_{L-1}$ after the feed-forward operations for gradient estimation during backward propagation.

### 2.2 ACTIVATION COMPRESSED TRAINING

It has been proved in existing work (Chen et al., 2020) that majority of memory (nearly 90%) is for caching activation maps during the training of DNNs. Following this direction, the activation compressed training (ACT) reduces the memory cost via real-time compressing the activation maps during the training. A typical ACT framework in existing work (Chakrabarti & Moseley, 2019) is shown in Figure 1 (b). Specifically, after the feed-forward operation of each layer $l$, activation map $\mathbf{H}_{l-1}$ is compressed into a representation for caching. The compression enables a significant reduction of memory compared with caching the original (exact) activation maps. During the backward pass of layer $l$, ACT decompresses the cached representation into $\hat{\mathbf{H}}_{l-1}$, and estimates the gradient by taking the reconstructed $\hat{\mathbf{H}}_{l-1}$ into Equation (2): $[\hat{\nabla}_{\mathbf{H}_{l-1}}, \hat{\nabla}_{\mathbf{W}_l}] = \text{backward}(\hat{\nabla}_{\mathbf{H}_l}, \hat{\mathbf{H}}_{l-1}, \mathbf{W}_l)$.

---

[3]We do not focus on the closed from the backward function, which is implemented by `torch.autograd`.

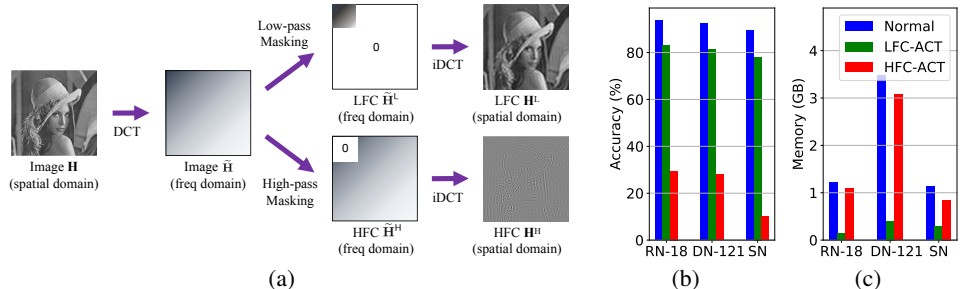

Figure 1: (a) Overall performance of DIVISION versus baseline methods. (b) Activation compressed training.

Figure 2: (a) Adopting DCT to estimate the low frequency component (LFC) and high frequency component (HFC) of an image. (b) Top-1 accuracy and (c) Memory cost of normal training, LFC-ACT and HFC-ACT, where RN-18, DN-121, and SN refer to the ResNet-18, DenseNet-121, and ShuffleNet-V2, respectively.

Even though the pipeline of compression and decompression is lossy, i.e. $\hat{\mathbf{H}}_l \neq \mathbf{H}_l$ for $1 \leq l \leq L$. It has been proved ACT can limit the reconstruction error flowing back to early layers and enables the training to approach an approximately optimal solution (Chen et al., 2021).

## 2.3 DISCRETE COSINE TRANSFORMATION

Discrete Cosine Transformation (DCT) projects the target data from the spatial domain to the frequency domain via the inner-production of the data and a collection of cosine functions with different frequency (Rao & Yip, 2014). We focus on the 2D-DCT in this work, where the target data is the input image and activation maps of DNNs. Specifically, for 2D-matrix data $\mathbf{H}$, the frequency-domain feature $\widetilde{\mathbf{H}}$ is estimated by $\widetilde{\mathbf{H}} = \mathrm{DCT}(\mathbf{H})$, where $\mathbf{H}$ and $\widetilde{\mathbf{H}}$ have the same shape of $N \times N$. and each of the element $\tilde{h}_{i,j}$ is given by

$$\tilde{h}_{i,j} = \sum_{m=0}^{N-1} \sum_{n=0}^{N-1} h_{m,n} \cos\left[\frac{\pi}{N}\left(m + \frac{1}{2}\right)i\right] \cos\left[\frac{\pi}{N}\left(n + \frac{1}{2}\right)j\right], \tag{3}$$

where $h_{m,n}$, $0 \leq m, n \leq N - 1$, are elements in the original matrix $\mathbf{H}$. During the training of DNNs, an image or activation map has the shape of $\mathrm{Minibatch} \times \mathrm{Channel} \times N \times N$. In this case, the frequency-domain feature is estimated via operating 2D-DCT for each $N \times N$ matrix in each channel.

With DCT, we could extract the low-frequency/high-frequency component (LFC/HFC) of an image or activation map, using a pipeline of low-pass/high-pass masking and inverse DCT, as shown in Figure 2. To be concrete, the estimation of LFC and HFC is given by

$$\mathbf{H}^{\mathsf{L}} = \mathrm{iDCT}(\widetilde{\mathbf{H}} \odot \mathbf{M}) \tag{4}$$

$$\mathbf{H}^{\mathsf{H}} = \mathrm{iDCT}(\widetilde{\mathbf{H}} \odot (\mathbf{1}_{N \times N} - \mathbf{M})), \tag{5}$$

where $\mathrm{iDCT}(\cdot)$ denotes the inverse DCT (Rao & Yip, 2014); $\mathbf{M} = [m_{i,j} | 1 \leq i, j \leq N]$ denotes an $N \times N$ low-pass mask satisfying $m_{i,j} = 1$ for $1 \leq i, j \leq W$ and $m_{i,j} = 0$ for other elements; and $\mathbf{1}_{N \times N} - \mathbf{M}$ indicates the high-pass mask. Intuitively, $\mathbf{H}^{\mathsf{L}}$ has $W^2$ non-zero float numbers in each channel, in contrast with $N^2 - W^2$ non-zero float numbers in each channel of $\mathbf{H}^{\mathsf{H}}$. Generally, we have $W \ll N$ in practical scenarios, e.g. $W/N = 0.1$ in Figure 2 (a). This indicates the HFC takes the majority of the memory cost in the caching of activation maps.

## 3 CONTRIBUTION OF LFC AND HFC TO BACKWARD PROPAGATION

In this section, we experimentally prove the LFC of activation maps has significantly more contribution to DNN backward propagation than the HFC. Moreover, our theoretical result indicates the LFC enables the estimated gradient to be bounded into a tighter range around the optimal value, leading to a more accurate learned model, which is consistent with the experimental results.

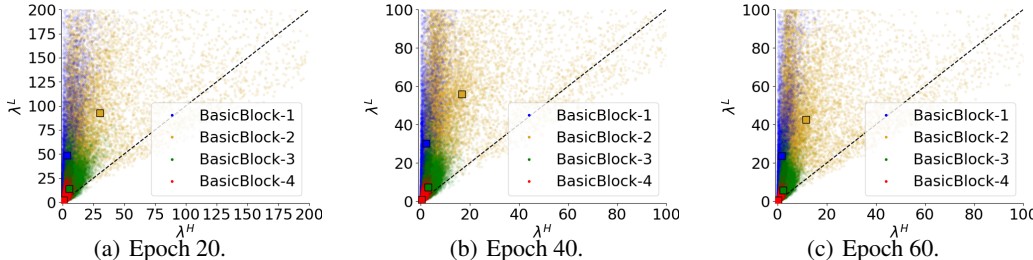

Figure 3: $\lambda_l^L = ||\widetilde{\mathbf{H}}_l \odot \mathbf{M}||_F$ versus $\lambda_l^H = ||\widetilde{\mathbf{H}}_l \odot (\mathbf{1}-\mathbf{M})||_F$ in the training (epoches 20, 40, and 60) of ResNet-18. $\mathbf{H}_l$ takes the activation maps of four BasicBlocks in ResNet-18; $\square$ indicates the mean values; $W = 0.5N$.

### 3.1 EXPERIMENTAL ANALYSIS

To study the individual contribution of LFC and HFC to DNN backward propagation, we design three training methods with different backward propagations: **LFC-ACT** takes LFC into the backward function as shown in Equation (6), where $\hat{\mathbf{H}}_l^L$ is estimated by Equations (4); **HFC-ACT** takes HFC into the backward function as given in Equation (7), where $\mathbf{H}_l^H$ is according to Equation (5); **Normal training** (for comparison) estimates the gradients by Equation (2).

$$[\hat{\nabla}_{\mathbf{H}_{l-1}}, \hat{\nabla}_{\mathbf{W}_l}] = \text{backward}(\hat{\nabla}_{\mathbf{H}_l}, \mathbf{H}_l^L, \mathbf{W}_l), \qquad \triangleright \text{LFC-ACT} \quad (6)$$

$$[\hat{\nabla}_{\mathbf{H}_{l-1}}, \hat{\nabla}_{\mathbf{W}_l}] = \text{backward}(\hat{\nabla}_{\mathbf{H}_l}, \mathbf{H}_l^H, \mathbf{W}_l). \qquad \triangleright \text{HFC-ACT} \quad (7)$$

We conduct the experiments on the CIFAR-10 dataset. The implementation details are given in Appendix A. The top-1 accuracy and memory cost of LFC-ACT, HFC-ACT, and normal training are shown in Figure 2 (b) and (c), respectively. Overall, we have the following observations:

- **Accuracy drop:** According to Figure 2 (b), HFC-ACT suffers from significantly more degradation of accuracy than LFC-ACT. This indicates *DNN backward propagation mainly utilizes the LFC of activation maps* during the training.

- **Memory cost:** According to Figure 2 (c), the storage of HFC requires significantly more memory than that of the LFC. i.e. *The storage of HFC consumes the majority of memory.*

To better understand the results of model accuracy, we theoretically prove the gradient for backward propagation is bounded into a tighter range around the optimal value in LFC-ACT. This enables LFC-ACT to learn a more accurate model than HFC-ACT.

### 3.2 THEORETICAL ANALYSIS

We theoretically analyze the gradient estimation error of LFC-ACT and HFC-ACT which adopt Equations (6) and (7) for backward propagation, respectively. Generally, for the LFC-ACT and HFC-ACT, let $\hat{\nabla}_{\mathbf{W}_l}^L$ and $\hat{\nabla}_{\mathbf{W}_l}^H$ denote the estimated gradient of layer $l$, respectively. In this way, $||\hat{\nabla}_{\mathbf{W}_l}^L - \nabla_{\mathbf{W}_l}||_F$[4] and $||\hat{\nabla}_{\mathbf{W}_l}^H - \nabla_{\mathbf{W}_l}||_F$ indicates the gradient estimation errors, taking the complete gradient $\nabla_{\mathbf{W}_l}$ as a reference. To compare the distortion of backward propagation in LFC-ACT and HFC-ACT, let $\text{GEB}_l^L$ and $\text{GEB}_l^H$ denote the gradient error upper bound (GEB), respectively, i.e. $||\hat{\nabla}_{\mathbf{W}_l}^L - \nabla_{\mathbf{W}_l}||_F \leq \text{GEB}_l^L$ and $||\hat{\nabla}_{\mathbf{W}_l}^H - \nabla_{\mathbf{W}_l}||_F \leq \text{GEB}_l^H$. Intuitively, higher GEB indicates less accurate backward propagation, leading to a less accurate model after training. To this end, we give Theorem 1 to compare $\text{GEB}_l^L$ and $\text{GEB}_l^H$, where a convolutional layer is considered. The proof is given in Appendix B. A similar analysis of GEB for a linear layer is provided in Appendix C.

**Theorem 1.** *During the backward pass of a convolutional layer $l$, $\text{GEB}_l^L$ and $\text{GEB}_l^H$ satisfy*

$$\text{GEB}_l^L - \text{GEB}_l^H = \left(\alpha_{l,l}||\mathbf{H}_{l-1}^T||_F + \beta_l\right)(\lambda_l^H - \lambda_l^L) + ||\mathbf{H}_{l-1}^T||_F \sum_{i=l+1}^{L} \alpha_{l,i}(\lambda_i^H - \lambda_i^L) \prod_{j=l}^{i-1} \gamma_j, \quad (8)$$

*where $\alpha_{l,i}, \beta_l, \gamma_l > 0$ for $1 \leq l, i \leq L$ depend on the model weights before backward propagation (given by Equations (24) in Appendix B); $\lambda_l^L = ||\widetilde{\mathbf{H}}_l \odot \mathbf{M}||_F$; $\lambda_l^H = ||\widetilde{\mathbf{H}}_l \odot (\mathbf{1}-\mathbf{M})||_F$; $\widetilde{\mathbf{H}}_l = \text{DCT}(\mathbf{H}_l)$; and $\mathbf{M}$ denotes the loss-pass mask given by Equation (4).*

Theorem 1 indicates the GEB difference depends on $\lambda_l^H - \lambda_l^L$ for $1 \leq l \leq L$ during the training. Following this direction, we estimate $\lambda_l^L$ and $\lambda_l^H$ via $\lambda_l^L = ||\widetilde{\mathbf{H}}_l \odot \mathbf{M}||_F$ and $\lambda_l^H = ||\widetilde{\mathbf{H}}_l \odot (\mathbf{1}-\mathbf{M})||_F$ during the training of ResNet-18 and [DenseNet-121](#) on the CIFAR-10 dataset. Specifically, $\mathbf{H}_l$ takes the

---

[4]The Frobenius norm of $n \times n$ matrix $\mathbf{A}$ is given by $||\mathbf{A}||_F = \sqrt{\sum_{i=1}^{n} \sum_{j=1}^{n} a_{ij}^2}$.

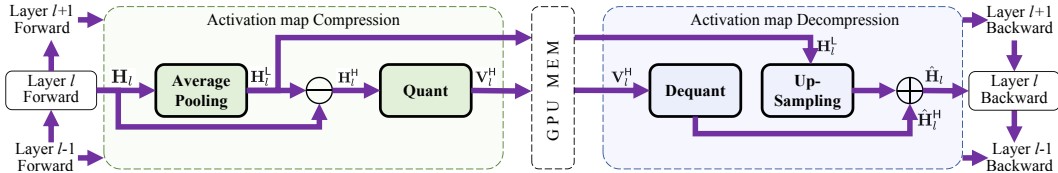

Figure 4: Framework of Dual Activation Precision Training.

activation maps of the BasicBlocks in ResNet-18, and Denseblocks in DenseNet-121. The estimation of $\lambda_l^{\mathsf{L}}$ and $\lambda_l^{\mathsf{H}}$ is based on the checkpoint of ResNet-18 in epochs 20, 40, and 60, and visualized in Figures 3 (a)-(c), respectively; and the results of DenseNet-121 are given in Appendix D. It is consistently observed that $\lambda_l^{\mathsf{L}} > \lambda_l^{\mathsf{H}}$ for different instances and layers. This leads to $\mathrm{GEB}_l^{\mathsf{L}} < \mathrm{GEB}_l^{\mathsf{H}}$ according to Theorem 1. Therefore, HFC-ACT suffers from a worse distortion of backward propagation during the training, eventually leading to less accurate learned model than LFC-ACT.

In this section, from both experimental and theoretical perspectives, we prove the HFC of activation maps has less contribution to backward propagation than LFC. However, according to Figure 2 (c), the HFC takes the majority of memory cost during the training. This indicates the HFC is highly redundant and compressible during the training. Following this direction, we propose DIVISION to compress the activation maps into a dual precision representation: *high-precision LFC* combined with *low-precision HFC*. On the one hand, both LFC and low-precision HFC requires much less memory to cache. On the other hand, removing the redundancy of HFC cannot cause much distortion of backward propagation. In this way, DIVISION enables effective compression of training memory without degradation of model accuracy.

## 4 DUAL ACTIVATION PRECISION TRAINING

We introduce the proposed Dual ActIVation PrecISION (DIVISION) in this section. The framework of DIVISION is shown in Figure 4. Specifically, after the feed-forward operation of each layer, DIVISION estimates the LFC and compresses the HFC into a low-precision copy such that the total memory cost is significantly decreased after the compression. Before the backward propagation of each layer, the low-precision HFC is decompressed and combined with LFC to reconstruct the activation map. The detailed compression and decompression are given as follows.

### 4.1 ACTIVATION MAP COMPRESSION

For compressing the activation map $\mathbf{H}_l$ of layer $l$, DIVISION estimates the LFC $\mathbf{H}_l^{\mathsf{L}}$ and HFC $\mathbf{H}_l^{\mathsf{H}}$ after the feed-forward operation. However, the high computational complexity of DCT prevents us from directly applying it to real-time algorithms. We thus give Theorem 2 to introduce a moving average operation that can approximate the loss-pass filter. The proof is given in Appendix E.

**Theorem 2.** *For any real-valued function $f(x)$ and its moving average $\bar{f}(x) = \frac{1}{2B} \int_x^{x+2B} f(t)\mathrm{d}t$, let $F(\omega)$ and $\overline{F}(\omega)$ denote the Fourier transformation of $f(x)$ and $\bar{f}(x)$, respectively. Generally, we have $\overline{F}(\omega) = H(\omega)F(\omega)$, where $|H(\omega)| = \left|\frac{\sin \omega B}{\omega B}\right|$.*

**Remark 1.** *The frequency response of $H(\omega)$ depends on its envelope function $\frac{1}{|\omega B|}$. Note that $\frac{1}{|\omega B|}$ decreases with $|\omega|$ such that $\frac{1}{|\omega B|} \to 0$ as $\omega \to \infty$. Hence, $H(\omega)$ is an approximate loss-pass filter.*

According to Remark 1, we approximate the LFC $\mathbf{H}_l^{\mathsf{L}}$ into the moving average of $\mathbf{H}_l$. Note that the average pooling operator provides efficient moving average. DIVISION adopts average pooling to estimate the LFC by $\mathbf{H}_l^{\mathsf{L}} = \mathrm{AveragePooling}(\mathbf{H}_l)$. The value of block-size and moving stride is a unified hyper-parameter $B$, which controls the memory of $\mathbf{H}_l^{\mathsf{L}}$[5]. Moreover, $\mathbf{H}_l^{\mathsf{L}}$ is cached in the format of `bfloat16` for saving the memory. In our experiments, we found $B = 8$ can provide representative LFC for backward propagation, where the memory cost of $\mathbf{H}_l^{\mathsf{L}}$ is only $0.8\%$ of $\mathbf{H}_l$.

To estimate the HFC, DIVISION calculates the residual value $\mathbf{H}_l^{\mathsf{H}} = \mathbf{H}_l - \mathrm{UpSampling}(\mathbf{H}_l^{\mathsf{L}})$, where the $\mathrm{UpSampling}(\cdot)$ enlarges $\mathbf{H}_l^{\mathsf{L}}$ to shape $\mathrm{Minibatch} \times \mathrm{Channel} \times N \times N$ via nearest interpolation. Then, DIVISION compress the $\mathbf{H}_l^{\mathsf{H}}$ into low-precision because it plays a less important role during the backward propagation but consumes most of the memory. Specifically, DIVISION adopts $Q$-bit

---

[5]For the case $N < B$, the pooling block-size and stride will be $N$ such that the shape of $\mathbf{H}_l^{\mathsf{L}}$ is $\mathrm{Minibatch} \times \mathrm{Channel} \times 1 \times 1$.

per-channel quantization[67] for the compression, where the bit-width $Q$ controls the precision and memory cost of HFC after the compression. Let $\mathbf{V}_l^{\mathsf{H}}$ denote a $Q$-bit integer matrix, as the low-precision representation of $\mathbf{H}_l^{\mathsf{H}}$. The detailed procedure of compressing $\mathbf{H}_l^{\mathsf{H}}$ into $\mathbf{V}_l^{\mathsf{H}}$ is given by

$$\mathbf{V}_l^{\mathsf{H}} = \mathrm{Quant}(\mathbf{H}_l^{\mathsf{H}}) = \lfloor \Delta_l^{-1}(\mathbf{H}_l^{\mathsf{H}} - \delta_l) \rceil, \tag{9}$$

where $\delta_l$ denotes the minimum element in $\mathbf{H}_l^{\mathsf{H}}$; $\Delta_l = (h_{\max} - \delta_l)/(2^Q - 1)$ denotes the quantization step; $h_{\max}$ denotes the maximum element in $\mathbf{H}_l^{\mathsf{H}}$; $\lfloor \bullet \rceil$ denotes the *stochastic rounding*[89] (Gupta et al., 2015); and $\delta_l$ and $\Delta_l$ are cached in the formate of `bfloat16` for saving memory. In this way, the memory cost of $(\mathbf{V}_l^{\mathsf{H}}, \delta_l, h_{\max})$ is $(N^2 Q/8 + 4)$ bytes per channel, in contrast with that of $\mathbf{H}_l$ being $4N^2$ bytes per channel. In our experiments, we found $Q = 2$ can provide enough representation for backward propagation, where the memory cost of $\mathbf{V}_l^{\mathsf{H}}$ is only $8.3\%$ of $\mathbf{H}_l$.

After the compression, as the representation of $\mathbf{H}_l$, the tuple of $(\mathbf{H}_l^{\mathsf{L}}, \mathbf{V}_l^{\mathsf{H}}, \Delta_l, \delta_l)$ is cached to the memory for reconstructing the activation maps during the backward pass.

## 4.2 ACTIVATION MAP DECOMPRESSION

During the backward pass, DIVISION adopts the cached tuples of $\{(\mathbf{H}_l^{\mathsf{L}}, \mathbf{V}_l^{\mathsf{H}}, \Delta_l, \delta_l) \mid 0 \leq l \leq L-1\}$ to reconstruct the activation map layer-by-layer. Specifically, for each layer $l$, DIVISION dequantizes the HFC via $\hat{\mathbf{H}}_l^{\mathsf{H}} = \Delta_l \mathbf{V}_l^{\mathsf{H}} + \delta_l$, which is the inverse process of Equation (9). Then, the activation map is reconstructed via

$$\hat{\mathbf{H}}_l = \mathrm{UpSampling}(\mathbf{H}_l^{\mathsf{L}}) + \hat{\mathbf{H}}_l^{\mathsf{H}}, \tag{10}$$

where $\mathrm{UpSampling}(\cdot)$ enlarges $\mathbf{H}_l^{\mathsf{L}}$ to shape $\mathrm{Minibatch} \times \mathrm{Channel} \times N \times N$ via nearest interpolation. After the decompression, DIVISION frees the caching of $(\mathbf{H}_l^{\mathsf{L}}, \mathbf{V}_l^{\mathsf{H}}, \Delta_l, \delta_l)$, and takes $\hat{\mathbf{H}}_l$ into $[\hat{\nabla}_{\mathbf{H}_{l-1}}, \hat{\nabla}_{\mathbf{W}_l}] = \mathrm{backward}(\hat{\nabla}_{\mathbf{H}_l}, \hat{\mathbf{H}}_{l-1}, \mathbf{W}_l)$ to estimate the gradient for backward propagation. Without loss of generality, 1D/3D activation maps are considered for DIVISION in Appendix F.

## 4.3 ALGORITHM OF DIVISION

Algorithm 1 demonstrates a mini-batch updating of DIVISION, which includes a forward pass and backward pass. During the forward pass of each layer, DIVISION first forwards the exact activation map to the next layer (line 2); then, estimates the LFC and HFC (line 3-4); after this, achieves the low precision copy of HFC (lines 5); finally caches the representation to the memory (line 6). During the backward pass of each layer, DIVISION first decompresses the HFC (line 10); then reconstructs the activation map (line 11); after this, estimates the gradients and updates the weights of layer $l$ (line 12); finally frees the caching of $(\mathbf{H}_{l-1}^{\mathsf{L}}, \mathbf{V}_{l-1}^{\mathsf{H}}, \Delta_{l-1}, \delta_{l-1})$ (line 13). For each mini-batch updating, the memory usage reaches the

| **Algorithm 1** Mini-batch updating of DIVISION |
| --- |
| **Input:** Mini-batch samples $\mathbf{x}$ and labels y. |
| **Output:** Weight and bias $\{\mathbf{W}_l, \mathbf{B}_l \mid 1 \leq l \leq L\}$. |
| 1: **for** *layer $l := 1$ to $L$* **do** |
| 2: $\quad \mathbf{H}_l = f(\mathbf{W}_l \mathbf{H}_{l-1} + \mathbf{B}_l)$ // $\mathbf{H}_0 = \mathbf{x}$ |
| 3: $\quad \mathbf{H}_{l-1}^{\mathsf{L}} = \mathrm{AveragePooling}(\mathbf{H}_{l-1})$ |
| 4: $\quad \mathbf{H}_{l-1}^{\mathsf{H}} = \mathbf{H}_{l-1} - \mathrm{UpSampling}(\mathbf{H}_{l-1}^{\mathsf{L}})$ |
| 5: $\quad \mathbf{V}_{l-1}^{\mathsf{H}}, \Delta_{l-1}, \delta_{l-1} = \mathrm{Quant}(\mathbf{H}_{l-1}^{\mathsf{H}})$ |
| 6: $\quad$ Cache $(\mathbf{H}_{l-1}^{\mathsf{L}}, \mathbf{V}_{l-1}^{\mathsf{H}}, \Delta_{l-1}, \delta_{l-1})$ |
| 7: **end for** |
| 8: Estimate the loss value and gradient $\hat{\nabla}_{\mathbf{H}_L}$. |
| 9: **for** *layer $l := L$ to 1* **do** |
| 10: $\quad \hat{\mathbf{H}}_{l-1}^{\mathsf{H}} = \mathrm{Dequant}(\mathbf{V}_{l-1}^{\mathsf{H}}, \Delta_{l-1}, \delta_{l-1})$ |
| 11: $\quad \hat{\mathbf{H}}_{l-1} = \mathrm{UpSampling}(\mathbf{H}_{l-1}^{\mathsf{L}}) + \hat{\mathbf{H}}_{l-1}^{\mathsf{H}}$ |
| 12: $\quad$ Estimate $[\hat{\nabla}_{\mathbf{H}_{l-1}}, \hat{\nabla}_{\mathbf{W}_l}]$ and update $\mathbf{W}_l$. |
| 13: $\quad$ Free $(\mathbf{H}_{l-1}^{\mathsf{L}}, \mathbf{V}_{l-1}^{\mathsf{H}}, \Delta_{l-1}, \delta_{l-1})$. |
| 14: **end for** |

maximum value after the forward pass (caching the representation of activation maps layer-by-layer), and reduces to the minimum value after the backward pass (freeing the cache layer-by-layer). Existing work (Chen et al., 2021) estimates the memory cost of activation maps by

$$\text{Memory Cost} = \text{Memory Utilization}_{\text{after forward}} - \text{Memory Utilization}_{\text{after backward}}, \tag{11}$$

where existing deep learning tools provide APIs[10] to estimate the memory utilization.

The theoretical compression rate $R$ of DIVISION is given in Appendix G, where general cases of convolutional neural networks and multi-layer perception are considered for the estimation. For the model architectures in our experiments, we have $R_{\text{ResNet-50}}, R_{\text{WRN-50-2}} \geq 10.35$.

---

[6] A fixed bit-width is adopted for the quantization of all layers to maximize the efficiency of data processing.

[7] Per-channel quantization is more efficient and light than per-group quantization in state-of-the-art work.

[8] $\lfloor x \rceil$ takes the value of $\lfloor x \rfloor$ with a probability of $x - \lfloor x \rfloor$ and takes $\lceil x \rceil$ with a probability of $\lceil x \rceil - x$.

[9] The stochastic rounding enables the quantization-dequantization pipeline to be unbiased, i.e. $\mathbb{E}[\mathbf{V}_l^{\mathsf{H}}] = \mathbf{H}_l^{\mathsf{H}}$.

[10] `torch.cuda.memory_allocated` returns the memory occupied by tensors in bytes.

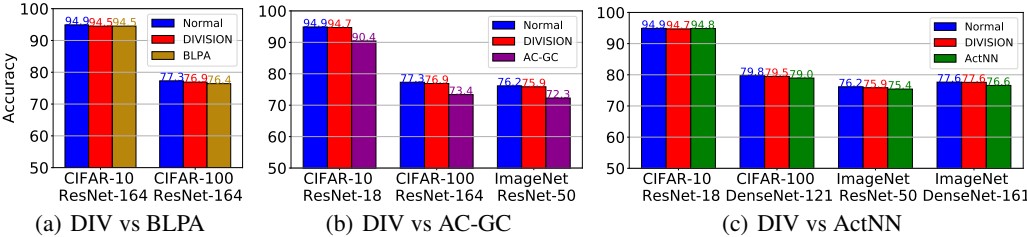

Figure 5: Top-1 accuracy (%) ↑ of Normal training, DIVISION, BLPA (a), AC-GC (b), and ActNN (c).

## 5 EVALUATION OF DIVISION

We conduct the experiments to evaluate DIVISION by answering the following research questions. **RQ1:** How does DIVISION perform compared with state-of-the-art baseline methods in terms of the model accuracy, memory cost, and training throughput? **RQ2:** Does the strategy of dual-precision compression contribute to DIVISION? **RQ3:** What is the effect of hyper-parameters on DIVISION?

The experiment setting including the datasets, baseline methods and DNN architectures is specified in Appendix H. The implementation details including the hyper-parameters of DIVISION and configuration of baseline methods are given in Appendix I. More experiments on MLPs, vision transformers, depthwise, and pointwise convolutional layers are given in Appendix K, L, and M.

### 5.1 EVALUATION BY MODEL ACCURACY (RQ1)

In this section, we evaluate the training methods in terms of model accuracy on the CIFAR-10, CIFAR-100 and ImageNet datasets. Specifically, DIVISION is compared with BLPA (Chakrabarti & Moseley, 2019), AC-GC (Evans & Aamodt, 2021) and ActNN (Chen et al., 2021) in Figure 5 (a)-(c), respectively, where different model architectures are considered. Here, Checkpoint and SWAP are not considered in this section because they are able to reduce the training memory without degradation of model accuracy. Overall, we have the following observations:

- **DIV vs Baseline Methods:** Compared with normal training, DIVISION achieves almost the same top-1 validation accuracy. In contrast, the baseline methods suffer from slightly higher validation error. This indicates DIVISION provides nearly loss-less compression of DNN training.
- **Flexibility of DIV:** DIVISION consistently achieves competitive model accuracy in the training of different architectures on different datasets. This indicates DIVISION is a flexible framework that can be applied to different scenarios.
- **Compressibility of HFC:** Note that DIVISION adopts a significantly high compression rate 12× for the HFC during the training, and achieves nearly loss-less accuracy. This result indicates the HFC of activation map is highly redundant and compressible during the training.

### 5.2 EVALUATION BY MEMORY COST (RQ1)

We evaluate the training methods in terms of the training memory cost on the ImageNet datasest, where the configuration of our computational infrastructure is given in Appendix R. Table 1 indicates the training memory cost and practical compression rate of DIVISION and baseline methods. Moreover, DIVISION is compared with the checkpoint strategy of Megatron-LM (Shoeybi et al., 2019) in Appendix O. Overall, we have the following observations:

- **DIV vs SWAP, Checkpoint & BLPA:** SWAP reduces the GPU memory cost merely by transferring the overhead from GPU to CPU, which is non-effective if considering the memory utilization of both GPU and CPU. Checkpoint shows considerable memory overhead because it caches some key activation maps to reconstruct other activation maps during backward pass. BLPA is less effective than DIVISION because it relies on at least 4-bit compression.
- **DIV vs AC-GC:** The practical memory cost of AC-GC should be greater than the values given in Table 1. AC-GC searches the bit-width from an initial maximum value, and finalizes with an optimal bit-width. Thus, the average memory cost should be greater than that in the last epoch.
- **DIV vs ActNN:** DIVISION has approximately the same memory cost as ActNN. Beyond the storage of 2-bit activation maps, DIVISION has overhead for caching the LFC; and ActNN spends almost equal overhead for storing the parameters of per-group quantization.

Table 1: Memory cost↓ and compression rate↑. *Total Mem* refers to total memory cost of weights, optimizer, data and activation maps. *Act Mem* refers to memory cost of activation maps. *OOM* refers to out of memory.

| Architecture | | ResNet-50 | | | | WRN-50-2 | | | |
|---|---|---|---|---|---|---|---|---|---|
| Batch-size | | 64 | 128 | 256 | 512 | 64 | 128 | 256 | 512 |
| Total Mem (GB) | Normal | 5.46 | 10.62 | 20.92 | *OOM* | 7.52 | 14.23 | 27.68 | *OOM* |
| | SWAP | 5.46 (1×) | 10.62 (1×) | 20.92 (1×) | *OOM* | 7.52 (1×) | 14.23 (1×) | 27.68 (1×) | *OOM* |
| | Checkpoint | 1.23 (4.4×) | 2.16 (4.9×) | 4.03 (5.2×) | 7.76 | 1.71 (4.4×) | 2.65 (5.4×) | 4.51 (6.1×) | 8.25 |
| | BLPA | 1.15 (4.7×) | 2.01 (5.3×) | 3.72 (5.6×) | 7.14 | 1.87 (4.0×) | 2.96 (4.8×) | 5.15 (5.4×) | 9.51 |
| | AC-GC | 1.80 (3.0×) | 3.31 (3.2×) | 6.31 (3.3×) | 12.33 | 2.72 (2.8×) | 4.66 (3.1×) | 8.53 (3.2×) | 16.27 |
| | ActNN | **0.81 (6.7×)** | **1.34 (7.9×)** | **2.39 (8.8×)** | **4.47** | **1.44 (5.2×)** | **2.09 (6.8×)** | **3.41 (8.1×)** | **6.03** |
| | DIVISION | 0.82 (6.7×) | 1.35 (7.9×) | 2.41 (8.7×) | 4.52 | 1.45 (5.2×) | 2.12 (6.7×) | 3.44 (8.0×) | 6.08 |
| Act. Mem (GB) | Normal | 5.14 | 10.25 | 20.48 | *OOM* | 6.70 | 13.38 | 26.75 | *OOM* |
| | SWAP | 5.14 (1×) | 10.25 (1×) | 20.48 (1×) | *OOM* | 6.70 (1×) | 13.38 (1×) | 26.75 (1×) | *OOM* |
| | Checkpoint | 0.90 (5.7×) | 1.80 (5.7×) | 3.59 (5.7×) | 7.18 | 0.90 (7.4×) | 1.80 (7.4×) | 3.59 (7.5×) | 7.18 |
| | BLPA | 0.82 (6.3×) | 1.64 (6.2×) | 3.28 (6.2×) | 6.56 | 1.06 (6.3×) | 2.11 (6.3×) | 4.22 (6.3×) | 8.44 |
| | AC-GC | 1.47 (3.5×) | 2.94 (3.5×) | 5.88 (3.5×) | 11.75 | 1.91 (3.5×) | 3.81 (3.5×) | 7.61 (3.5×) | 15.20 |
| | ActNN | **0.49 (10.5×)** | **0.97 (10.6×)** | **1.94 (10.6×)** | **3.89** | **0.62 (10.8×)** | **1.25 (10.7×)** | **2.49 (10.7×)** | **4.97** |
| | DIVISION | 0.49 (10.5×) | 0.99 (10.4×) | 1.97 (10.4×) | 3.94 | 0.64 (10.5×) | 1.27 (10.5×) | 2.52 (10.6×) | 5.02 |

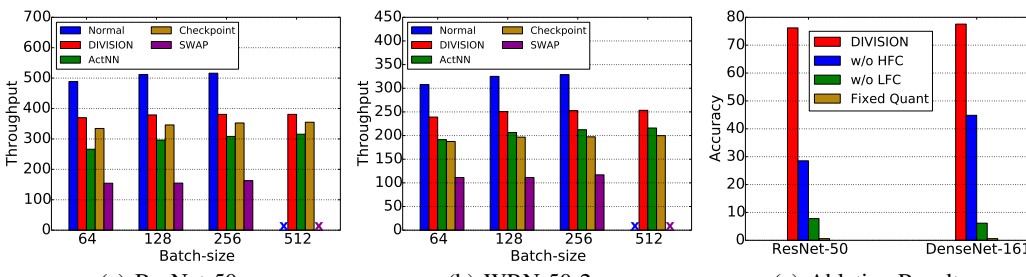

(a) ResNet-50.    (b) WRN-50-2.    (c) Ablation Results.

Figure 6: Training throughput ↑ of (a) Resnet-50 and (b) WRN-50-2 on the ImageNet dataset, where ✖ indicates out of memory. (c) Top-1 validation accuracy (%) ↑ of DIVISION, DIVISION w/o HFC, DIVISION w/o LFC and fixed bit-width quantization on the ImageNet dataset.

- **Act. Maps:** For normal training, the caching of activation maps takes the majority of memory cost (>90%, growing with the mini-batch size), which is consistent with our discussion in Section 1.

- **Compression rate:** The activation map compression rate of DIVISION is consistently with the theoretical results ($R_{\text{ResNet-50}}, R_{\text{WRN-50-2}} \geq 10.35$, see Appendix G), which is not influenced by the mini-batch size. Moreover, the overall compression rate grows with the mini-batch size.

## 5.3 EVALUATION BY TRAINING THROUGHPUT (RQ1)

We now evaluate the training methods in terms of the training throughput on the Imagenet dataset. Generally, the throughput indicates the running speed of a training method via counting the average number of data samples processed per second. The throughput is given by $\frac{\text{Mini-batch Size}}{T_{\text{batch}}}$, where $T_{\text{batch}}$ denotes the time consumption of single mini-batch updating. Each method is combined with the automatic mixed precision (AMP)[11] to speed up the training. Figures 6 (a) and (b) show the average throughput of 20 times of mini-batch updating. Overall, we have the following observations:

- **Reason for Time overhead:** Compared with normal training, the time overhead of DIVISION comes from the estimation of LFC and compression of HFC. In ActNN, the overhead mainly comes from the the dynamic bit-width allocation and activation map quantization. In Checkpoint, it comes from replaying the forward process of inter-media layers. In SWAP, the overhead mainly derives from the communication cost between the CPUs and GPUs.

- **DIV vs ActNN:** DIVISION shows higher throughput than ActNN as a result of more simplified data compression. To be concrete, DIVISION adopts average-pooling to extract the LFC, and a fixed bit-width per-channel quantization to compress the HFC. In contrast, ActNN relies on searching optimal bit-width to match different samples, and per-group quantization based on the searched bit-width. ActNN has more complex processing, which leads to its lower throughput.

- **DIV vs SWAP:** SWAP is less efficient than ACT-based methods (DIVISION and ActNN), which indicates the CPU-GPU communication cost is larger than the cost of activation map processing.

[11] https://pytorch.org/docs/stable/amp.html

### 5.4 Effect of Dual Precision Strategy (RQ2)

To study the effect of our proposed dual precision strategy, DIVISION is compared with three training methods: **DIVISION w/o HFC**: Merely caching the high-precision LFC for back-propagation. **DIVISION w/o LFC**: Merely caching the low-precision HFC for back-propagation. **Fixed Quant**: Compressing the activation maps using a fixed bit-width quantization. The experiments are conducted on the ImageNet dataset using the hyper-parameters given in Appendix S. More experiments of Fixed Quant with different bit-width are given in Appendix P. The model accuracy are given in Figure 6 (c). Overall, we have the following insights:

- **LFC & Low Precision HFC:** Removing either HFC or LFC from DIVISION, the training converges to far lower levels of accuracy. This indicates both the LFC and low precision HFC of activation maps are necessary for leading the training to converge to an optimal solution.

- **Benifits of Dual Precision:** The fixed bit-width quantization fails to converge to an optimal solution. This indicates the noise caused by the fixed bit-width quantization can terribly disturb the back-propagation. DIVISION solves this problem by combining a high-precision LFC and a fixed bit-width quantization for compressing the activation maps.

### 5.5 Hyper-parameter Tuning for DIVISION (RQ3)

We study the effect of hyper-parameters $B$ (block-size) and $Q$ (bit-width) on the accuracy and compression rate. Specifically, we adopt DIVISION to train ResNet-18 on the CIFAR-10 dataset with $B \in \{8, 12, 18\}$ and $Q \in \{2, 4, 8\}$. The accuracy versus compression rate is shown in Figure 7 (a). Overall, we have the following insights:

(a) Effect of Hyper-parameters.

(b) Effect of AMP.

- **Effect of $Q$:** The accuracy is stable (consistently nearly $95\%$) when reducing the precision-level of HFC ($Q$ reduces from 8 to 2). This indicates DIVISION only requires approximate values of HFC during backward propagation.

- **Effect of $B$:** Lower-precision LFC for backward propagation leads to significant degradation of accuracy (as $B$ grows from 8 to 18). This is because DIVISION relies on a high-precision LFC to reconstruct the activation maps for backward propagation.

- **Optimal Setting:** DIVISION has optimal accuracy-compression trade-off taking $B = 8$ and $Q = 2$, where the degradation of accuracy is less than $0.4\%$. According to more empirical studies in Appendix N, $B = 8$ and $Q = 2$ can be a default setting effective for most of model architectures and datasets.

Figure 7: (a) Top-1 Accuracy and compression rate of DIVISION in different settings. (b) Top-1 accuracy and training throughput of DIVISION w/ and w/o AMP.

Note that normal training can be accelerated by the automatic mixed precision (AMP) (Micikevicius et al., 2017) without loss of accuracy. We study whether AMP can speed up DIVISION without loss of accuracy. Specifically, we follow the setting of DIVISION $B = 8$, $Q = 2$ to train ResNet-18 on the CIFAR-10 dataset. The accuracy and training throughput of DIVISION w/ and w/o AMP are shown in Figure 7 (b). More experiments with different mini-batch size are given in Appendix Q. It is observed that AMP can significantly speed up the DIVISION when $\mathrm{MiniBatch\text{-}size} \geq 256$ without loss of model accuracy. This indicates DIVISION has the potential to be applied to scenarios where both time and memory are limited.

## 6 Conclusion

In this work, we propose a simple framework of activation compressed training. Our framework is motivated by an instructive observation: *DNN backward propagation mainly depends on the LFC of the activation maps, while the majority of memory is for the storage of HFC during the training.* This indicates back-propagation mainly utilizes the LFC to estimate the gradient, while the HFC is highly redundant and compressible. Following this direction, our proposed DIVISION compresses the activation maps into dual precision representations: high-precision LFC and low-precision HFC, according to their contributions to the back-propagation. This dual precision compression can significantly reduce the memory cost of activation maps without disturbing the training.

Different from the existing work of ACT, DIVISION is a simple and transparent framework, where the simplicity enables efficient compression and decompression; and transparency allows us to understand the compressible (HFC) and non-compressible factors (LFC) during DNN training. To this end, we hope our work could provide some inspiration for the compression of DNN training.

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

APPENDIX

## A    IMPLEMENTATION DETAILS OF SECTION 3

We give the details of the experiment in Section 3. Without loss of generality, the experiment is conducted on the CIFAR-10 dataset using ResNet-18, DenseNet-121 and ShuffleNet-V2. During the backward propagation of normal training, the gradient of each layer $l$ is estimated by

$$[\hat{\nabla}_{\mathbf{H}_{l-1}}, \hat{\nabla}_{\mathbf{W}_l}] = \text{backward}(\hat{\nabla}_{\mathbf{H}_l}, \mathbf{H}_l, \mathbf{W}_l) \qquad (12)$$

For LFC-ACT, the gradient is estimated by

$$[\hat{\nabla}_{\mathbf{H}_{l-1}}, \hat{\nabla}_{\mathbf{W}_l}] = \text{backward}(\hat{\nabla}_{\mathbf{H}_l}, \mathbf{H}_l^{\mathsf{L}}, \mathbf{W}_l), \qquad (13)$$

where HFC-ACT denotes the HFC of $\mathbf{H}_l$; for HFC-ACT, the gradient is estimated by

$$[\hat{\nabla}_{\mathbf{H}_{l-1}}, \hat{\nabla}_{\mathbf{W}_l}] = \text{backward}(\hat{\nabla}_{\mathbf{H}_l}, \mathbf{H}_l^{\mathsf{H}}, \mathbf{W}_l), \qquad (14)$$

where $\mathbf{H}_l^{\mathsf{H}}$ denotes the HFC of $\mathbf{H}_l$. Note that Equations (13) and (14) causes the distortion of backward propagation in LFC-ACT and HFC-ACT, respectively. The objective of this experiment is to investigate whether this distortion of back-propagation may be powerful enough to lead training to a non-optimal solution. The hyper-parameter setting of the training is given in Table 2.

Table 2: Hyper-parameter setting.

| Architecture | ResNet-18 | DenseNet-121 | ShuffleNet-V2 |
|---|---|---|---|
| Epoch | 100 | 100 | 100 |
| Batch-size | 256 | 256 | 256 |
| Initial LR | 0.1 | 0.1 | 0.1 |
| LR scheduler | Step LR | Step LR | Step LR |
| Weight-decay | 0.0005 | 0.0005 | 0.0005 |
| Optimizer | SGD | SGD | SGD |
| SGD Momentum | 0.9 | 0.9 | 0.9 |
| Ratio of LFC ($W/N$) | 0.3 | 0.3 | 0.5 |

## B    PROOF OF THEOREM 1

We prove Theorem 1 in this section.

**Theorem 1** *During the backward pass of a convolutional layer $l$, $\text{GEB}_l^{\mathsf{L}}$ and $\text{GEB}_l^{\mathsf{H}}$ satisfy*

$$\text{GEB}_l^{\mathsf{L}} - \text{GEB}_l^{\mathsf{H}} = \left(\alpha_{l,l}||\mathbf{H}_{l-1}^{\mathsf{T}}||_F + \beta_l\right)(\lambda_l^{\mathsf{H}} - \lambda_l^{\mathsf{L}}) + ||\mathbf{H}_{l-1}^{\mathsf{T}}||_F \sum_{i=l+1}^{L} \alpha_{l,i}(\lambda_i^{\mathsf{H}} - \lambda_i^{\mathsf{L}}) \prod_{j=l}^{i-1} \gamma_j, \qquad (15)$$

*where $\alpha_{l,i}, \beta_l, \gamma_l > 0$ for $1 \leq l, i \leq L$ are given by Equation (24); $\lambda_l^{\mathsf{L}} = ||\widetilde{\mathbf{H}}_l \odot \mathbf{M}||_F$; $\lambda_l^{\mathsf{H}} = ||\widetilde{\mathbf{H}}_l \odot (\mathbf{1} - \mathbf{M})||_F$; $\widetilde{\mathbf{H}}_l = \text{DCT}(\mathbf{H}_l)$; and $\mathbf{M}$ denotes the loss-pass mask given by Equation (4).*

*Proof.* For simplicity of derivation, we study the case with a single input channel and output channel number. In this case, $\mathbf{H}_l$ and $\mathbf{W}_l$ are 2-D matrix for each layer $l$, where $1 \leq l \leq L$. The backward propagation of a convolutional layer is given by

$$\begin{aligned} \hat{\nabla}_{\mathbf{Z}_l} &= \hat{\nabla}_{\mathbf{Z}_{l+1}} * \mathbf{W}_{l+1}^{\text{rot}} \odot \sigma'(\hat{\mathbf{Z}}_l), \\ \hat{\nabla}_{\mathbf{W}_l} &= \hat{\nabla}_{\mathbf{Z}_l} * \hat{\mathbf{H}}_{l-1}^{\mathsf{T}}, \end{aligned} \qquad (16)$$

where $*$ denotes a convolutional operation; $\hat{\mathbf{Z}}_l = \mathbf{W}_l * \hat{\mathbf{H}}_{l-1} + b_l$; $b_l$ denotes the bias of layer $l$; and $\mathbf{W}_l^{\text{rot}}$ denotes to rotate $\mathbf{W}_l$ by $180°$. The case of multiple input and output channels can be proved in an analogous way, which is omitted in this work.

According to Equation (16), we have the gradient of $\mathbf{Z}_l$ given by

$$\hat{\nabla}_{\mathbf{Z}_l} - \nabla_{\mathbf{Z}_l}$$
$$= \hat{\nabla}_{\mathbf{Z}_{l+1}} * \mathbf{W}_{l+1}^{\text{rot}} \odot \sigma'(\hat{\mathbf{Z}}_l) - \nabla_{\mathbf{Z}_{l+1}} * \mathbf{W}_{l+1}^{\text{rot}} \odot \sigma'(\mathbf{Z}_l),$$
$$= \hat{\nabla}_{\mathbf{Z}_{l+1}} * \mathbf{W}_{l+1}^{\text{rot}} \odot \sigma'(\hat{\mathbf{Z}}_l) - \hat{\nabla}_{\mathbf{Z}_{l+1}} * \mathbf{W}_{l+1}^{\text{rot}} \odot \sigma'(\mathbf{Z}_l) + \hat{\nabla}_{\mathbf{Z}_{l+1}} * \mathbf{W}_{l+1}^{\text{rot}} \odot \sigma'(\mathbf{Z}_l) - \nabla_{\mathbf{Z}_{l+1}} * \mathbf{W}_{l+1}^{\text{rot}} \odot \sigma'(\mathbf{Z}_l),$$
$$= \hat{\nabla}_{\mathbf{Z}_{l+1}} * \mathbf{W}_{l+1}^{\text{rot}} \odot [\sigma'(\hat{\mathbf{Z}}_l) - \sigma'(\mathbf{Z}_l)] + (\hat{\nabla}_{\mathbf{Z}_{l+1}} - \nabla_{\mathbf{Z}_{l+1}}) * \mathbf{W}_{l+1}^{\text{rot}} \odot \sigma'(\mathbf{Z}_l). \tag{17}$$

For the activation functions $\mathrm{ReLu}(\cdot)$, $\mathrm{LeakyReLu}(\cdot)$, $\mathrm{Sigmoid}(\cdot)$, $\mathrm{Tanh}(\cdot)$ and $\mathrm{SoftPlus}(\cdot)$, the gradient $\sigma'(\cdot)$ satisfies $|\sigma''(\cdot)| \leq 1$ in the differentable domains. Note that we have $||\mathbf{W}_l * \mathbf{H}_{l-1}||_F \leq (K_l + N_l - 1)||\mathbf{W}_l||_F ||\mathbf{H}_{l-1}||_F$ according to Corollary 1. $||\sigma'(\hat{\mathbf{Z}}_l) - \sigma'(\mathbf{Z}_l)||_F$ satisfies

$$||\sigma'(\hat{\mathbf{Z}}_l) - \sigma'(\mathbf{Z}_l)||_F \leq ||\hat{\mathbf{Z}}_l - \mathbf{Z}_l||_F \leq (K_l + N_l - 1)||\hat{\mathbf{H}}_{l-1} - \mathbf{H}_{l-1}||_F ||\mathbf{W}_l'||_F, \tag{18}$$

where $K_l$ and $N_l$ denote the size of convolutional kernel $\mathbf{W}_l$ and activation map $\mathbf{H}_l$ in layer $l$, respectively. After taking Equation (18) into Equation (17), we have

$$||\hat{\nabla}_{\mathbf{Z}_l} - \nabla_{\mathbf{Z}_l}||_F \leq (K_l + N_l - 1)||\hat{\nabla}_{\mathbf{Z}_{l+1}}||_F ||\mathbf{W}_{l+1}^{\text{rot}}||_F ||\sigma'(\hat{\mathbf{Z}}_l) - \sigma'(\mathbf{Z}_l)||_F$$
$$+ (K_l + N_l - 1)||\hat{\nabla}_{\mathbf{Z}_{l+1}} - \nabla_{\mathbf{Z}_{l+1}}||_F ||\mathbf{W}_{l+1}^{\text{rot}}||_F ||\sigma'(\mathbf{Z}_l)||_F,$$
$$= (K_l + N_l - 1)^2 ||\hat{\nabla}_{\mathbf{Z}_{l+1}}||_F ||\mathbf{W}_{l+1}^{\text{rot}}||_F ||\hat{\mathbf{H}}_{l-1} - \mathbf{H}_{l-1}||_F ||\mathbf{W}_l'||_F$$
$$+ (K_l + N_l - 1)||\hat{\nabla}_{\mathbf{Z}_{l+1}} - \nabla_{\mathbf{Z}_{l+1}}||_F ||\mathbf{W}_{l+1}^{\text{rot}}||_F ||\sigma'(\mathbf{Z}_l)||_F,$$
$$= \eta_l ||\hat{\mathbf{H}}_{l-1} - \mathbf{H}_{l-1}||_F + \gamma_l ||\hat{\nabla}_{\mathbf{Z}_{l+1}} - \nabla_{\mathbf{Z}_{l+1}}||_F, \tag{19}$$

where $\eta_l$ and $\gamma_l$ are given by

$$\eta_l = (K_l + N_l - 1)^2 ||\hat{\nabla}_{\mathbf{Z}_{l+1}}||_F ||\mathbf{W}_{l+1}||_F ||\mathbf{W}_l'||_F;$$
$$\gamma_l = (K_l + N_l - 1)||\mathbf{W}_{l+1}||_F ||\sigma'(\mathbf{Z}_l)||_F; \tag{20}$$

the value $\eta_l$ and $\gamma_l$ depend on the model weight before backward propagation, which is constant with respect to the gradient. Iterate Equation (19) until $l = L$ where $||\hat{\nabla}_{\mathbf{Z}_L} - \nabla_{\mathbf{Z}_L}||_F \leq \eta_L ||\hat{\mathbf{H}}_{L-1} - \mathbf{H}_{L-1}||_F$. In this way, we have

$$||\hat{\nabla}_{\mathbf{Z}_l} - \nabla_{\mathbf{Z}_l}||_F \leq \eta_l ||\hat{\mathbf{H}}_{l-1} - \mathbf{H}_{l-1}||_F + \sum_{i=l+1}^{L} \eta_i ||\hat{\mathbf{H}}_{i-1} - \mathbf{H}_{i-1}||_F \prod_{j=l}^{i-1} \gamma_j. \tag{21}$$

According to Equation (16), we have the gradient of $\mathbf{W}_l$ given by

$$\hat{\nabla}_{\mathbf{W}_l} - \nabla_{\mathbf{W}_l} = \hat{\nabla}_{\mathbf{Z}_l} * \hat{\mathbf{H}}_{l-1}^{\mathsf{T}} - \nabla_{\mathbf{Z}_l} * \mathbf{H}_{l-1}^{\mathsf{T}},$$
$$= \hat{\nabla}_{\mathbf{Z}_l} * \hat{\mathbf{H}}_{l-1}^{\mathsf{T}} - \hat{\nabla}_{\mathbf{Z}_l} * \mathbf{H}_{l-1}^{\mathsf{T}} + \hat{\nabla}_{\mathbf{Z}_l} * \mathbf{H}_{l-1}^{\mathsf{T}} - \nabla_{\mathbf{Z}_l} * \mathbf{H}_{l-1}^{\mathsf{T}},$$
$$= \hat{\nabla}_{\mathbf{Z}_l} * (\hat{\mathbf{H}}_{l-1}^{\mathsf{T}} - \mathbf{H}_{l-1}^{\mathsf{T}}) + (\hat{\nabla}_{\mathbf{Z}_l} - \nabla_{\mathbf{Z}_l}) * \mathbf{H}_{l-1}^{\mathsf{T}}. \tag{22}$$

Taking Equation (21) into Equation (22), we have

$$||\hat{\nabla}_{\mathbf{W}_l} - \nabla_{\mathbf{W}_l}||_F$$
$$\leq (K_l + N_l - 1)||\hat{\nabla}_{\mathbf{Z}_l}||_F ||\hat{\mathbf{H}}_{l-1}^{\mathsf{T}} - \mathbf{H}_{l-1}^{\mathsf{T}}||_F + (K_l + N_l - 1)||\hat{\nabla}_{\mathbf{Z}_l} - \nabla_{\mathbf{Z}_l}||_F ||\mathbf{H}_{l-1}^{\mathsf{T}}||_F,$$
$$\leq (K_l + N_l - 1)\left( ||\hat{\nabla}_{\mathbf{Z}_l}||_F ||\hat{\mathbf{H}}_{l-1}^{\mathsf{T}} - \mathbf{H}_{l-1}^{\mathsf{T}}||_F + ||\mathbf{H}_{l-1}^{\mathsf{T}}||_F \left[ \eta_l ||\hat{\mathbf{H}}_{l-1} - \mathbf{H}_{l-1}||_F + \sum_{i=l+1}^{L} \eta_i ||\hat{\mathbf{H}}_{i-1} - \mathbf{H}_{i-1}||_F \prod_{j=l}^{i-1} \gamma_j \right] \right)$$
$$= (K_l + N_l - 1)\left[ (||\hat{\nabla}_{\mathbf{Z}_l}||_F + \eta_l ||\mathbf{H}_{l-1}^{\mathsf{T}}||_F) ||\hat{\mathbf{H}}_{l-1}^{\mathsf{T}} - \mathbf{H}_{l-1}^{\mathsf{T}}||_F + ||\mathbf{H}_{l-1}^{\mathsf{T}}||_F \sum_{i=l+1}^{L} \eta_i ||\hat{\mathbf{H}}_{i-1} - \mathbf{H}_{i-1}||_F \prod_{j=l}^{i-1} \gamma_j \right],$$
$$= \left( \beta_l + \alpha_{l,l} ||\mathbf{H}_{l-1}^{\mathsf{T}}||_F \right) ||\hat{\mathbf{H}}_{l-1}^{\mathsf{T}} - \mathbf{H}_{l-1}^{\mathsf{T}}||_F + ||\mathbf{H}_{l-1}^{\mathsf{T}}||_F \sum_{i=l+1}^{L} \alpha_{l,i} ||\hat{\mathbf{H}}_{i-1}^{\mathsf{T}} - \mathbf{H}_{i-1}^{\mathsf{T}}||_F \prod_{j=l}^{i-1} \gamma_j, \tag{23}$$

where

$$
\begin{aligned}
\alpha_{l,i} &= (K_l + N_l - 1)(K_i + N_i - 1)^2 ||\hat{\nabla}_{\mathbf{Z}_{i+1}}||_F ||\mathbf{W}_{i+1}||_F ||\mathbf{W}'_i||_F; \\
\beta_l &= (K_l + N_l - 1)||\hat{\nabla}_{\mathbf{Z}_l}||_F; \\
\gamma_l &= (K_l + N_l - 1)||\mathbf{W}_{l+1}||_F ||\sigma'(\mathbf{Z}_l)||_F;
\end{aligned}
\tag{24}
$$

$K_l$ and $N_l$ denote the size of convolutional kernel $\mathbf{W}_l$ and activation map $\mathbf{H}_l$ in layer $l$, respectively.

During the LFC-ACT and HFC-ACT trainings, the activation map of a convolutional layer satisfies

$$
||\mathbf{H}_l - \mathbf{H}_l^{\mathsf{L}}||_F = ||\widetilde{\mathbf{H}}_l - \widetilde{\mathbf{H}}_l^{\mathsf{L}}||_F = ||\widetilde{\mathbf{H}}_l \odot (\mathbf{1} - \mathbf{M})||_F \triangleq \lambda_l^{\mathsf{H}},
\tag{25}
$$

$$
||\mathbf{H}_l - \mathbf{H}_l^{\mathsf{H}}||_F = ||\widetilde{\mathbf{H}}_l - \widetilde{\mathbf{H}}_l^{\mathsf{H}}||_F = ||\widetilde{\mathbf{H}}_l \odot \mathbf{M}||_F \triangleq \lambda_l^{\mathsf{L}}.
\tag{26}
$$

Taking Equations (25) and (26) into (23), we have $\mathrm{GEB}_l^{\mathsf{L}}$ and $\mathrm{GEB}_l^{\mathsf{H}}$ of a convolutional layer by

$$
||\hat{\nabla}_{\mathbf{W}_l} - \nabla_{\mathbf{W}_l}^{\mathsf{L}}||_F \leq \left(\alpha_{l,l}||\mathbf{H}_{l-1}^{\mathsf{T}}||_F + \beta_l\right)\lambda_l^{\mathsf{H}} + ||\mathbf{H}_{l-1}^{\mathsf{T}}||_F \sum_{i=l+1}^{L} \alpha_{l,i} \lambda_i^{\mathsf{H}} \prod_{j=l}^{i-1} \gamma_j \triangleq \mathrm{GEB}_l^{\mathsf{L}},
\tag{27}
$$

$$
||\hat{\nabla}_{\mathbf{W}_l} - \nabla_{\mathbf{W}_l}^{\mathsf{H}}||_F \leq \left(\alpha_{l,l}||\mathbf{H}_{l-1}^{\mathsf{T}}||_F + \beta_l\right)\lambda_l^{\mathsf{L}} + ||\mathbf{H}_{l-1}^{\mathsf{T}}||_F \sum_{i=l+1}^{L} \alpha_{l,i} \lambda_i^{\mathsf{L}} \prod_{j=l}^{i-1} \gamma_j \triangleq \mathrm{GEB}_l^{\mathsf{H}}.
\tag{28}
$$

Given the expression of $\mathrm{GEB}_l^{\mathsf{L}}$ and $\mathrm{GEB}_l^{\mathsf{H}}$ by Equations (27) and (28), respectively, we have the GEB for a convolutional layer given by

$$
\mathrm{GEB}_l^{\mathsf{L}} - \mathrm{GEB}_l^{\mathsf{H}} = \left(\alpha_{l,l}||\mathbf{H}_{l-1}^{\mathsf{T}}||_F + \beta_l\right)(\lambda_l^{\mathsf{H}} - \lambda_l^{\mathsf{L}}) + ||\mathbf{H}_{l-1}^{\mathsf{T}}||_F \sum_{i=l+1}^{L} \alpha_{l,i}(\lambda_i^{\mathsf{H}} - \lambda_i^{\mathsf{L}}) \prod_{j=l}^{i-1} \gamma_j.
$$

$\square$

**Corollary 1.** *For a $K \times K$ convolutional kernel and a $N \times N$ square matrix $\mathbf{H}$, we have the*

$$
||\mathbf{W} * \mathbf{H}||_F \leq (K + N - 1)||\mathbf{W}||_F ||\mathbf{H}||_F
\tag{29}
$$

*Proof.* According to the relations between convolutional operation and Discrete Fourier Transformation (Sundararajan, 2001), $\mathbf{W} * \mathbf{H}$ satisfies

$$
\mathrm{FFT}(\mathbf{W} * \mathbf{H}) = \mathrm{FFT}(\mathrm{ZP}(\mathbf{W})) \odot \mathrm{FFT}(\mathrm{ZP}(\mathbf{H})),
\tag{30}
$$

where $\mathrm{FFT}(\cdot)$ denotes the discrete Fourier transformation; $\mathrm{ZP}(\mathbf{W})$ denotes zero-padding $\mathbf{W}$ into a $(K + N - 1) \times (K + N - 1)$ matrix. According to the Parseval's theorem (Diniz et al., 2010), $\mathrm{FFT}(\mathrm{ZP}(\mathbf{W}))$ and $\mathrm{FFT}(\mathrm{ZP}(\mathbf{H}))$ and $\mathrm{FFT}(\mathbf{W} * \mathbf{H})$ satisfy

$$
\begin{aligned}
||\mathrm{FFT}(\mathrm{ZP}(\mathbf{W}))||_F &= (K + N - 1)||\mathbf{W}||_F, \\
||\mathrm{FFT}(\mathrm{ZP}(\mathbf{H}))||_F &= (K + N - 1)||\mathbf{H}||_F, \\
||\mathrm{FFT}(\mathbf{W} * \mathbf{H})||_F &= (K + N - 1)||\mathbf{W} * \mathbf{H}||_F.
\end{aligned}
\tag{31}
$$

Taking $||\mathbf{A}_1 \odot \mathbf{A}_2||_F \leq ||\mathbf{A}_1||_F ||\mathbf{A}_2||_F$ into Equation (31), we have

$$
\mathrm{FFT}(\mathrm{ZP}(\mathbf{W})) \odot \mathrm{FFT}(\mathrm{ZP}(\mathbf{H})) \leq ||\mathrm{FFT}(\mathbf{W})||_F ||\mathrm{FFT}(\mathbf{H})||_F
\tag{32}
$$

Taking Equation (31) into Equation (32), we have

$$
\begin{aligned}
(K + N - 1)||\mathbf{W} * \mathbf{H}||_F &= ||\mathrm{FFT}(\mathbf{W}) \odot \mathrm{FFT}(\mathbf{H})||_F \\
&\leq ||\mathrm{FFT}(\mathbf{W})||_F ||\mathrm{FFT}(\mathbf{H})||_F \\
&= (K + N - 1)||\mathbf{W}||_F (K + N - 1)||\mathbf{H}||_F
\end{aligned}
$$

$\square$

## C GRADIENT ERROR BOUND (GEB) OF A LINEAR LAYER

We give the Gradient Error upper Bound (GEB) of a linear layer and proof in this section.

**Theorem 1B.** *During the backward pass of a linear layer $l$,* $\mathrm{GEB}_l^{\mathsf{L}}$ *and* $\mathrm{GEB}_l^{\mathsf{H}}$ *satisfy*

$$\mathrm{GEB}_l^{\mathsf{L}}-\mathrm{GEB}_l^{\mathsf{H}} = \left(\alpha_l||\mathbf{H}_{l-1}^{\mathsf{T}}||_F+\beta_l\right)(\lambda_l^{\mathsf{H}}-\lambda_l^{\mathsf{L}})+||\mathbf{H}_{l-1}^{\mathsf{T}}||_F\sum_{i=l+1}^{L}\alpha_i(\lambda_i^{\mathsf{H}}-\lambda_i^{\mathsf{L}})\prod_{j=l}^{i-1}\gamma_j, \quad (33)$$

*where $\alpha_l, \beta_l, \gamma_l > 0$ for $1\leq l\leq L$ are given by Equation (42); $\lambda_l^{\mathsf{L}}=||\widetilde{\mathbf{H}}_l\odot\mathbf{M}||_F$; $\lambda_l^{\mathsf{H}}=||\widetilde{\mathbf{H}}_l\odot(\mathbf{1}-\mathbf{M})||_F$; $\widetilde{\mathbf{H}}_l = \mathrm{DCT}(\mathbf{H}_l)$; and $\mathbf{M}$ denotes the 1-D loss-pass mask.*

*Proof.* For simplicity of derivation, we consider the case MiniBatch=1. In this case, $\mathbf{H}_l$ is a vector; and $\mathbf{W}_l$ is a 2-D matrix, for $1\leq l\leq L$. The backward propagation of a linear layer is given by

$$\begin{aligned}
\hat{\nabla}_{\mathbf{Z}_l} &= (\mathbf{W}_{l+1}\hat{\nabla}_{\mathbf{Z}_{l+1}}) \odot \sigma'(\hat{\mathbf{Z}}_l), \\
\hat{\nabla}_{\mathbf{W}_l} &= \hat{\nabla}_{\mathbf{Z}_l}\hat{\mathbf{H}}_{l-1}^{\mathsf{T}},
\end{aligned} \quad (34)$$

where $\hat{\mathbf{Z}}_l = \mathbf{W}_l^{\mathsf{T}}\hat{\mathbf{H}}_{l-1} + b_l$; and $b_l$ denotes the bias of layer $l$. The case of MiniBatch $\geq 2$ can be proved in an analogous way, which is omitted in this work.

According to Equation (34), we have the gradient of $\mathbf{Z}_l$ given by

$$\begin{aligned}
&\hat{\nabla}_{\mathbf{Z}_l} - \nabla_{\mathbf{Z}_l} \\
&= \mathbf{W}_{l+1}\hat{\nabla}_{\mathbf{Z}_{l+1}} \odot \sigma'(\hat{\mathbf{Z}}_l) - \mathbf{W}_{l+1}\nabla_{\mathbf{Z}_{l+1}} \odot \sigma'(\mathbf{Z}_l), \\
&= \mathbf{W}_{l+1}\hat{\nabla}_{\mathbf{Z}_{l+1}}\odot\sigma'(\hat{\mathbf{Z}}_l) - \mathbf{W}_{l+1}\hat{\nabla}_{\mathbf{Z}_{l+1}}\odot\sigma'(\mathbf{Z}_l) + \mathbf{W}_{l+1}\hat{\nabla}_{\mathbf{Z}_{l+1}}\odot\sigma'(\mathbf{Z}_l) - \mathbf{W}_{l+1}\nabla_{\mathbf{Z}_{l+1}}\odot\sigma'(\mathbf{Z}_l), \\
&= \mathbf{W}_{l+1}\hat{\nabla}_{\mathbf{Z}_{l+1}} \odot [\sigma'(\hat{\mathbf{Z}}_l) - \sigma'(\mathbf{Z}_l)] + (\hat{\nabla}_{\mathbf{Z}_{l+1}} - \mathbf{W}_{l+1}\nabla_{\mathbf{Z}_{l+1}}) \odot \sigma'(\mathbf{Z}_l). \quad (35)
\end{aligned}$$

For activation functions $\mathrm{ReLu}(\cdot)$, $\mathrm{LeakyReLu}(\cdot)$, $\mathrm{Sigmoid}(\cdot)$, $\mathrm{Tanh}(\cdot)$ and $\mathrm{SoftPlus}(\cdot)$, the gradient $\sigma'(\cdot)$ satisfies $|\sigma''(\cdot)| \leq 1$ in each differentiable domain. Combined with Cauchy–Schwarz inequality $||\mathbf{A}_1\mathbf{A}_2||_F \leq ||\mathbf{A}_1||_F||\mathbf{A}_2||_F$ (Horn & Johnson, 2012), we have

$$||\sigma'(\hat{\mathbf{Z}}_l) - \sigma'(\mathbf{Z}_l)||_F \leq ||\hat{\mathbf{Z}}_l - \mathbf{Z}_l||_F \leq ||\mathbf{W}_l||_F||\hat{\mathbf{H}}_{l-1} - \mathbf{H}_{l-1}||_F. \quad (36)$$

According to inequality $||\mathbf{A}_1\odot\mathbf{A}_2||_F \leq ||\mathbf{A}_1||_F||\mathbf{A}_2||_F$ (Horn & Johnson, 2012), we have the upper bound of $||\hat{\nabla}_{\mathbf{Z}_l} - \nabla_{\mathbf{Z}_l}||_F$ given by

$$\begin{aligned}
&||\hat{\nabla}_{\mathbf{Z}_l} - \nabla_{\mathbf{Z}_l}||_F \\
&\leq ||\mathbf{W}_{l+1}||_F||\hat{\nabla}_{\mathbf{Z}_{l+1}}||_F||\sigma'(\hat{\mathbf{Z}}_l) - \sigma'(\mathbf{Z}_l)||_F + ||\mathbf{W}_{l+1}||_F||\hat{\nabla}_{\mathbf{Z}_{l+1}} - \nabla_{\mathbf{Z}_{l+1}}||_F||\sigma'(\mathbf{Z}_l)||_F, \\
&= ||\mathbf{W}_{l+1}||_F||\hat{\nabla}_{\mathbf{Z}_{l+1}}||_F||\hat{\mathbf{H}}_{l-1}-\mathbf{H}_{l-1}||_F||\mathbf{W}_l'||_F + ||\mathbf{W}_{l+1}||_F||\hat{\nabla}_{\mathbf{Z}_{l+1}}-\nabla_{\mathbf{Z}_{l+1}}||_F||\sigma'(\mathbf{Z}_l)||_F, \\
&= \alpha_l||\hat{\mathbf{H}}_{l-1} - \mathbf{H}_{l-1}||_F + \gamma_l||\hat{\nabla}_{\mathbf{Z}_{l+1}} - \nabla_{\mathbf{Z}_{l+1}}||_F, \quad (37)
\end{aligned}$$

where $\alpha_l$ and $\gamma_l$ are given by

$$\begin{aligned}
\alpha_l &= ||\mathbf{W}_{l+1}||_F||\hat{\nabla}_{\mathbf{Z}_{l+1}}||_F||\mathbf{W}_l'||_F; \\
\gamma_l &= ||\mathbf{W}_{l+1}||_F||\sigma'(\mathbf{Z}_l)||_F;
\end{aligned} \quad (38)$$

the value $\alpha_l$ and $\gamma_l$ depend on the model weight before backward propagation, which are constant with respect to the gradient. Iterate Equation (37) until $l = L$ where $||\hat{\nabla}_{\mathbf{Z}_L} - \nabla_{\mathbf{Z}_L}||_F \leq \alpha_l||\hat{\mathbf{H}}_{L-1}-\mathbf{H}_{L-1}||_F$. In such a manner, we have

$$||\hat{\nabla}_{\mathbf{Z}_l} - \nabla_{\mathbf{Z}_l}||_F \leq \alpha_l||\hat{\mathbf{H}}_{l-1}-\mathbf{H}_{l-1}||_F + \sum_{i=l+1}^{L}\alpha_i||\hat{\mathbf{H}}_{i-1}-\mathbf{H}_{i-1}||_F\prod_{j=l}^{i-1}\gamma_j. \quad (39)$$

According to Equation (34), we have the gradient of $\mathbf{W}_l$ given by

$$\begin{aligned}
\hat{\nabla}_{\mathbf{W}_l} - \nabla_{\mathbf{W}_l} &= \hat{\nabla}_{\mathbf{Z}_l}\hat{\mathbf{H}}_{l-1}^{\mathsf{T}} - \nabla_{\mathbf{Z}_l}\mathbf{H}_{l-1}^{\mathsf{T}}, \\
&= \hat{\nabla}_{\mathbf{Z}_l}\hat{\mathbf{H}}_{l-1}^{\mathsf{T}} - \hat{\nabla}_{\mathbf{Z}_l}\mathbf{H}_{l-1}^{\mathsf{T}} + \hat{\nabla}_{\mathbf{Z}_l}\mathbf{H}_{l-1}^{\mathsf{T}} - \nabla_{\mathbf{Z}_l}\mathbf{H}_{l-1}^{\mathsf{T}}, \\
&= \hat{\nabla}_{\mathbf{Z}_l}(\hat{\mathbf{H}}_{l-1}^{\mathsf{T}} - \mathbf{H}_{l-1}^{\mathsf{T}}) + (\hat{\nabla}_{\mathbf{Z}_l} - \nabla_{\mathbf{Z}_l})\mathbf{H}_{l-1}^{\mathsf{T}}. \quad (40)
\end{aligned}$$

Taking Equation (39) into Equation (40), we have

$$||\hat{\nabla}_{\mathbf{W}_l} - \nabla_{\mathbf{W}_l}||_F$$
$$\leq ||\hat{\nabla}_{\mathbf{z}_l}||_F||\hat{\mathbf{H}}_{l-1}^\mathsf{T} - \mathbf{H}_{l-1}^\mathsf{T}||_F + ||\hat{\nabla}_{\mathbf{z}_l} - \nabla_{\mathbf{z}_l}||_F||\mathbf{H}_{l-1}^\mathsf{T}||_F,$$
$$\leq ||\hat{\nabla}_{\mathbf{z}_l}||_F||\hat{\mathbf{H}}_{l-1}^\mathsf{T} - \mathbf{H}_{l-1}^\mathsf{T}||_F + ||\mathbf{H}_{l-1}^\mathsf{T}||_F\left[\alpha_l||\hat{\mathbf{H}}_{l-1} - \mathbf{H}_{l-1}||_F + \sum_{i=l+1}^{L}\alpha_i||\hat{\mathbf{H}}_{i-1} - \mathbf{H}_{i-1}||_F\prod_{j=l}^{i-1}\gamma_j\right],$$
$$= \left(\beta_l + \alpha_l||\mathbf{H}_{l-1}^\mathsf{T}||_F\right)||\hat{\mathbf{H}}_{l-1}^\mathsf{T} - \mathbf{H}_{l-1}^\mathsf{T}||_F + ||\mathbf{H}_{l-1}^\mathsf{T}||_F\sum_{i=l+1}^{L}\alpha_i||\hat{\mathbf{H}}_{i-1}^\mathsf{T} - \mathbf{H}_{i-1}^\mathsf{T}||_F\prod_{j=l}^{i-1}\gamma_j, \qquad (41)$$

where $\beta_l$ is given by

$$\begin{aligned}\alpha_l &= ||\mathbf{W}_{l+1}||_F||\hat{\nabla}_{\mathbf{z}_{l+1}}||_F||\mathbf{W}_l'||_F;\\ \beta_l &= ||\hat{\nabla}_{\mathbf{z}_l}||_F;\\ \gamma_l &= ||\mathbf{W}_{l+1}||_F||\sigma'(\mathbf{Z}_l)||_F.\end{aligned} \qquad (42)$$

During the LFC-ACT and HFC-ACT trainings, the activation map of a linear layer satisfies

$$||\mathbf{H}_l - \mathbf{H}_l^\mathsf{L}||_F = ||\widetilde{\mathbf{H}}_l - \widetilde{\mathbf{H}}_l^\mathsf{L}||_F = ||\widetilde{\mathbf{H}}_l \odot (\mathbf{1} - \mathbf{M})||_F \triangleq \lambda_l^\mathsf{H}, \qquad (43)$$
$$||\mathbf{H}_l - \mathbf{H}_l^\mathsf{H}||_F = ||\widetilde{\mathbf{H}}_l - \widetilde{\mathbf{H}}_l^\mathsf{H}||_F = ||\widetilde{\mathbf{H}}_l \odot \mathbf{M}||_F \triangleq \lambda_l^\mathsf{L}. \qquad (44)$$

Taking Equations (43) and (44) into (41), we have the $\text{GEB}_l^\mathsf{L}$ and $\text{GEB}_l^\mathsf{H}$ of a linear layer given by

$$||\hat{\nabla}_{\mathbf{W}_l} - \nabla_{\mathbf{W}_l}^\mathsf{L}||_F \leq \left(\alpha_l||\mathbf{H}_{l-1}^\mathsf{T}||_F + \beta_l\right)\lambda_l^\mathsf{H} + ||\mathbf{H}_{l-1}^\mathsf{T}||_F\sum_{i=l+1}^{L}\alpha_i\lambda_i^\mathsf{H}\prod_{j=l}^{i-1}\gamma_j \triangleq \text{GEB}_l^\mathsf{L}, \qquad (45)$$

$$||\hat{\nabla}_{\mathbf{W}_l} - \nabla_{\mathbf{W}_l}^\mathsf{H}||_F \leq \left(||\alpha_l||\mathbf{H}_{l-1}^\mathsf{T}||_F + \beta_l\right)\lambda_l^\mathsf{L} + ||\mathbf{H}_{l-1}^\mathsf{T}||_F\sum_{i=l+1}^{L}\alpha_i\lambda_i^\mathsf{L}\prod_{j=l}^{i-1}\gamma_j \triangleq \text{GEB}_l^\mathsf{H}. \qquad (46)$$

Given the expression of $\text{GEB}_l^\mathsf{L}$ and $\text{GEB}_l^\mathsf{H}$ by Equations (45) and (46), we have the GEB difference for a linear layer given by

$$\text{GEB}_l^\mathsf{L} - \text{GEB}_l^\mathsf{H} = \left(||\alpha_l||\mathbf{H}_{l-1}^\mathsf{T}||_F + \beta_l\right)(\lambda_l^\mathsf{H} - \lambda_l^\mathsf{L}) + ||\mathbf{H}_{l-1}^\mathsf{T}||_F\sum_{i=l+1}^{L}\alpha_i(\lambda_i^\mathsf{H} - \lambda_i^\mathsf{L})\prod_{j=l}^{i-1}\gamma_j.$$

$$\square$$

## D  $\lambda_l^L$ VERSUS $\lambda_l^H$ IN DENSENET-121

A further experiment is conducted to study whether $\lambda_l^L > \lambda_l^H$ can be guaranteed for DenseNet-121. Specifically, the values of $\lambda_l^L$ and $\lambda_l^H$ in the training (epoches 20, 40, and 60) of DenseNet-121 are given in Tables 3 and 4, where the low-pass $M$ mask satisfies $W/N = 0.1$ and $W/N = 0.2$, respectively; and $\lambda_l^L$ and $\lambda_l^H$ are estimated based on the input activation maps of the four DenseBlocks. It is consistently observed that $\lambda_l^L > \lambda_l^H$ for $W/N = 0.1$ and $W/N = 0.2$ in different training epochs. This indicates our proposed Theorem 1 holds without loss of generality.

## E  PROOF OF THEOREM 2

We give the proof of Theorem 2 in this section.

**Theorem 2.** *For any real-valued function $f(x)$ and its moving average $\bar{f}(x) = \frac{1}{2B}\int_x^{x+2B} f(t)\mathrm{d}t$, let $F(\omega)$ and $\overline{F}(\omega)$ denote the Fourier transformation (Madisetti, 1997) of $f(x)$ and $\bar{f}(x)$, respectively. Generally, we have $\overline{F}(\omega) = H(\omega)F(\omega)$, where $|H(\omega)| = \left|\frac{\sin \omega B}{\omega B}\right|$.*

Table 3: $W/N = 0.1$.

| Epoch | 20 | 40 | 60 | Average $\frac{\lambda^L}{\lambda^H}$ |
|---|---|---|---|---|
| DenseBlock-1 | $\lambda_l^L = 298.281$ $\lambda_l^H = 218.605$ | $\lambda_l^L = 184.913$ $\lambda_l^H = 138.069$ | $\lambda_l^L = 142.668$ $\lambda_l^H = 104.755$ | 1.36 |
| DenseBlock-2 | $\lambda_l^L = 3.245$ $\lambda_l^H = 1.713$ | $\lambda_l^L = 1.284$ $\lambda_l^H = 0.689$ | $\lambda_l^L = 0.687$ $\lambda_l^H = 0.372$ | 1.87 |
| DenseBlock-3 | $\lambda_l^L = 0.387$ $\lambda_l^H = 0.260$ | $\lambda_l^L = 0.160$ $\lambda_l^H = 0.086$ | $\lambda_l^L = 0.084$ $\lambda_l^H = 0.048$ | 1.70 |
| DenseBlock-4 | $\lambda_l^L = 0.062$ $\lambda_l^H = 0.009$ | $\lambda_l^L = 0.011$ $\lambda_l^H = 0.001$ | $\lambda_l^L = 0.006$ $\lambda_l^H = 0.001$ | 7.56 |
| Average $\lambda_l^L/\lambda_l^H$ | 2.95 | 3.12 | 3.30 | 3.12 |

Table 4: $W/N = 0.2$.

| Epoch | 20 | 40 | 60 | Average $\lambda_l^L/\lambda_l^H$ |
|---|---|---|---|---|
| DenseBlock-1 | $\lambda_l^L = 362.672$ $\lambda_l^H = 154.214$ | $\lambda_l^L = 225.543$ $\lambda_l^H = 97.439$ | $\lambda_l^L = 173.595$ $\lambda_l^H = 73.828$ | 2.34 |
| DenseBlock-2 | $\lambda_l^L = 3.632$ $\lambda_l^H = 1.326$ | $\lambda_l^L = 1.440$ $\lambda_l^H = 0.533$ | $\lambda_l^L = 0.774$ $\lambda_l^H = 0.285$ | 2.72 |
| DenseBlock-3 | $\lambda_l^L = 0.445$ $\lambda_l^H = 0.202$ | $\lambda_l^L = 0.179$ $\lambda_l^H = 0.067$ | $\lambda_l^L = 0.095$ $\lambda_l^H = 0.037$ | 2.49 |
| DenseBlock-4 | $\lambda_l^L = 0.062$ $\lambda_l^H = 0.009$ | $\lambda_l^L = 0.011$ $\lambda_l^H = 0.001$ | $\lambda_l^L = 0.006$ $\lambda_l^H = 0.001$ | 7.56 |
| Average $\lambda_l^L/\lambda_l^H$ | 3.58 | 3.78 | 3.97 | 3.78 |

*Proof.* We adopt the limit operator to reformulate $\bar{f}(x)$ into

$$\bar{f}(x) = \frac{1}{2B} \int_x^{x+2B} f(t)\mathrm{d}t = \frac{1}{2B} \lim_{N\to\infty} \sum_{n=0}^{N-1} \frac{2B}{N} f(x + \frac{2Bn}{N}) = \lim_{N\to\infty} \sum_{n=0}^{N-1} \frac{1}{N} f(x + \frac{2Bn}{N}) \quad (47)$$

Taking Equation (47) into the Fourier Transform of $\bar{f}(x)$, we have

$$
\begin{aligned}
F'(\omega) &= \int_{-\infty}^{\infty} \bar{f}(x)e^{-i\omega x}\mathrm{d}x = \int_{-\infty}^{\infty} \frac{1}{N} \lim_{N\to\infty} \sum_{n=0}^{N-1} f(x + \frac{2Bn}{N})e^{-i\omega x}\mathrm{d}x \\
&= \lim_{N\to\infty} \frac{1}{N} \sum_{n=0}^{N-1} \int_{-\infty}^{\infty} f(x + \frac{2Bn}{N})e^{-i\omega x}\mathrm{d}x \\
&= \lim_{N\to\infty} \frac{1}{N} \sum_{n=0}^{N-1} e^{i\omega\frac{2Bn}{N}} \int_{-\infty}^{\infty} f(x)e^{-i\omega x}\mathrm{d}x = F(\omega) \lim_{N\to\infty} \frac{1}{N} \sum_{n=0}^{N-1} e^{i\omega\frac{2Bn}{N}} \\
&= F(\omega)(1 - e^{i\omega 2B}) \lim_{N\to\infty} \frac{1}{N(1 - e^{i\omega\frac{2B}{N}})} = F(\omega)\frac{1 - e^{i\omega 2B}}{-i\omega 2B},
\end{aligned}
\quad (48)
$$

where $i$ denotes the imaginary unit.

Let $H(\omega) = \frac{1-e^{i\omega 2B}}{-i\omega 2B}$. The magnitude of $H(\omega)$ is given by

$$
\begin{aligned}
\left| H(\omega) \right| &= \frac{|1 - \cos\omega 2B + i\sin\omega 2B|}{|\omega 2B|} = \frac{\sqrt{(1 - \cos\omega 2B)^2 + \sin^2\omega 2B|}}{|\omega 2B} \\
&= \frac{\sqrt{4\sin^4\omega B + 4\sin^2\omega B\cos^2\omega B}}{|\omega 2B|} = \frac{\sqrt{4\sin^2\omega B(\sin^2\omega B + \cos^2\omega B)}}{|\omega 2B|} \\
&= \left| \frac{\sin\omega B}{\omega B} \right|
\end{aligned}
\quad (49)
$$

□

## F COMPRESSION OF 1D AND 3D ACTIVATION MAPS BY DIVISION

We give more details about DIVISION considering 1D, 2D and 3D activation maps in this section.

### F.1 ACTIVATION MAP COMPRESSION

DIVISION adopts average pooling to estimate the LFC by $\mathbf{H}_l^{\mathsf{L}} = \text{AveragePooling}(\mathbf{H}_l)$. The value of block-size and moving stride is a unified hyper-parameter $B$, which controls the memory of $\mathbf{H}_l^{\mathsf{L}}$. The average pooling of 1D, 2D and 3D activation maps are considered as follows,

$$\text{Minibatch} \times \text{Channel} \times N \xrightarrow{\text{AveragePooling1}D} \text{Minibatch} \times \text{Channel} \times \lfloor N/B \rfloor \qquad (50)$$

$$\text{Minibatch} \times \text{Channel} \times N \times N \xrightarrow{\text{AveragePooling2}D} \text{Minibatch} \times \text{Channel} \times \lfloor N/B \rfloor \times \lfloor N/B \rfloor$$

$$\text{Minibatch} \times \text{Channel} \times N \times N \times N \xrightarrow{\text{AveragePooling3}D} \text{Minibatch} \times \text{Channel} \times \lfloor N/B \rfloor \times \lfloor N/B \rfloor \times \lfloor N/B \rfloor$$

To estimate the HFC, DIVISION calculates the residual value $\mathbf{H}_l^{\mathsf{H}} = \mathbf{H}_l - \text{UpSampling}(\mathbf{H}_l^{\mathsf{L}})$, where the UpSampling($\cdot$) enlarges $\mathbf{H}_l^{\mathsf{L}}$ to the shape of $\mathbf{H}_l$ via nearest interpolation. The up sampling of 1D, 2D and 3D activation maps are considered as follows,

$$\text{Minibatch} \times \text{Channel} \times \lfloor N/B \rfloor \xrightarrow{\text{UpSampling1}D} \text{Minibatch} \times \text{Channel} \times N \quad (51)$$

$$\text{Minibatch} \times \text{Channel} \times \lfloor N/B \rfloor \times \lfloor N/B \rfloor \xrightarrow{\text{UpSampling2}D} \text{Minibatch} \times \text{Channel} \times N \times N$$

$$\text{Minibatch} \times \text{Channel} \times \lfloor N/B \rfloor \times \lfloor N/B \rfloor \times \lfloor N/B \rfloor \xrightarrow{\text{UpSampling3}D} \text{Minibatch} \times \text{Channel} \times N \times N \times N$$

Then, DIVISION adopts $Q$-bit per-channel quantization for the compression, where the bit-width $Q$ controls the precision and memory cost of HFC after the compression. Let $\mathbf{V}_l^{\mathsf{H}}$ denote a $Q$-bit integer matrix, as the low-precision representation of $\mathbf{H}_l^{\mathsf{H}}$. The detailed procedure of compressing $\mathbf{H}_l^{\mathsf{H}}$ into $\mathbf{V}_l^{\mathsf{H}}$ is given by

$$\mathbf{V}_l^{\mathsf{H}} = \text{Quant}(\mathbf{H}_l^{\mathsf{H}}) = \lfloor \Delta_l^{-1}(\mathbf{H}_l^{\mathsf{H}} - \delta_l) \rceil, \qquad (52)$$

where $\delta_l$ denotes the minimum element in $\mathbf{H}_l^{\mathsf{H}}$; $\Delta_l = (h_{\max} - \delta_l)/(2^Q - 1)$ denotes the quantization step; $h_{\max}$ denotes the maximum element in $\mathbf{H}_l^{\mathsf{H}}$; $\lfloor \bullet \rceil$ denotes the stochastic rounding.

After the compression, as the representation of $\mathbf{H}_l$, the tuple of $(\mathbf{H}_l^{\mathsf{L}}, \mathbf{V}_l^{\mathsf{H}}, \Delta_l, \delta_l)$ is cached to the memory for reconstructing the activation maps during the backward pass.

### F.2 ACTIVATION MAP DECOMPRESSION

During the backward pass, DIVISION adopts the cached tuples of $\{(\mathbf{H}_l^{\mathsf{L}}, \mathbf{V}_l^{\mathsf{H}}, \Delta_l, \delta_l) \mid 0 \leq l \leq L-1\}$ to reconstruct the activation map layer-by-layer. Specifically, for each layer $l$, DIVISION dequantizes the HFC via $\hat{\mathbf{H}}_l^{\mathsf{H}} = \Delta_l \mathbf{V}_l^{\mathsf{H}} + \delta_l$, which is the inverse process of Equation (52). Then, the activation map is reconstructed via $\hat{\mathbf{H}}_l = \text{UpSampling}(\mathbf{H}_l^{\mathsf{L}}) + \hat{\mathbf{H}}_l^{\mathsf{H}}$, where UpSampling($\cdot$) enlarges $\mathbf{H}_l^{\mathsf{L}}$ to the shape of $\mathbf{H}_l$ via nearest interpolation. The cases of 1D, 2D and 3D activation maps are considered in Equation (51).

After the decompression, DIVISION frees the caching of $(\mathbf{H}_l^{\mathsf{L}}, \mathbf{V}_l^{\mathsf{H}}, \Delta_l, \delta_l)$, and takes $\hat{\mathbf{H}}_l$ into $[\hat{\nabla}_{\mathbf{H}_{l-1}}, \hat{\nabla}_{\mathbf{W}_l}] = \text{backward}(\hat{\nabla}_{\mathbf{H}_l}, \hat{\mathbf{H}}_{l-1}, \mathbf{W}_l)$ to estimate the gradient for backward propagation.

## G THEORETICAL COMPRESSION RATE OF DIVISION

The compression rate of DIVISION is estimated in this section. A general case of convolutional neural networks (CNN) and multi-layer perceptron (MLP) are considered for the estimation.

### G.1 COMPRESSION RATE OF CNN TRAINING

Without loss of generality, we estimate the compression rate for a block of convolutional layer (`conv`), batch normalization layer (`BN`) and Relu activation. Most of existing backbones purely stacks `conv-BN-Relu` blocks (He et al., 2016; Huang et al., 2017; Szegedy et al., 2015; Tan & Le, 2019; Simonyan & Zisserman, 2014), which makes our estimated compression rate hold in practice. Generally, the compression rate is defined as the memory reduction ratio after the compression. To be concrete, let $\text{Minibatch} \times \text{Channel} \times N \times N$ denote the shape of activaition maps for a `conv-BN-Relu` block; given the block-size $B$ and bit-width $Q$, DIVISION has the compression rate of activation maps given by Theorem 3A.

**Theorem 3A.** *DIVISION has average activation map compression rate for a `conv-BN-Relu` block given by*

$$R = \frac{Mem\ of\ \mathbf{H}}{Mem\ of\ (\mathbf{H^L}, \mathbf{V^H}, \Delta, \delta)} = \frac{9}{\frac{4}{\min\{B^2,N^2\}} + \frac{Q}{4} + \frac{8}{N^2} + \frac{1}{8}}, \tag{53}$$

*where* $\text{Minibatch} \times \text{Channel} \times N \times N$ *is the shape of activation map* $\boldsymbol{H}_l$*; $B$ denotes the block-size of LFC average pooling; and $Q$ denotes the bit-width of HFC quantization.*

*Proof.* For each mini-batch updating of normal training, a `conv`-layer or `BN`-layer caches $N^2$`float32` $\times$ 4byte/`float32` $= 4N^2$byte activation map; a Relu operator caches $N^2$`int8` $\times$ 1byte/`int8` $= N^2$byte activation map. For each mini-batch updating of DIVISION, a `conv`-layer or `BN`-layer caches $\frac{N^2}{\min\{B^2,N^2\}}$`bfloat16` $\times$ 2byte/`bfloat16` $= \frac{2N^2}{\min\{B^2,N^2\}}$byte LFC; and $QN^2$bit $\times \frac{1}{8}$bit/byte $= \frac{Q}{8}N^2$byte HFC; and spends 2`bfloat16` $\times$ 2byte/`bfloat16` $= 4$byte for $\Delta_l$ and $\delta_l$. Moreover, a Relu operator caches $N^2$bit $\times \frac{1}{8}$byte/bit $= \frac{N^2}{8}$byte activation map. Therefore, the average activation map compression rate of a `conv-BN-Relu` block is given by

$$R = \frac{4N^2 \times 2 + N^2}{\left(\frac{2N^2}{\min\{B^2,N^2\}} + \frac{Q}{8}N^2 + 4\right) \times 2 + \frac{1}{8}N^2} = \frac{9}{\frac{4}{\min\{B^2,N^2\}} + \frac{Q}{4} + \frac{8}{N^2} + \frac{1}{8}}. \tag{54}$$

$\square$

A higher compression rate indicates more effective compression. It is observed that the compression rate grows with $B$ and $N$, and decreases with $Q$. In our experiments, we found $B = 8$ and $Q = 2$ can provide loss-less model accuracy. In this condition, the shape of activation maps satisfies $N \geq 7$ for ResNet-50 and WRN-50-2 on the ImageNet dataset (He et al., 2016). According to Equation (53), we have $R_{\text{ResNet-50}}, R_{\text{WRN-50-2}} \geq 10.35$.

### G.2 COMPRESSION RATE OF MLP TRAINING

We estimate the compression rate for a `linear-Relu` block in Theorem 3B. An MLP simply stacks multiple `linear-Relu` blocks, such that our estimated compression rate holds for MLP models.

**Theorem 3B.** *DIVISION has average activation map compression rate for a `linear-Relu` block given by*

$$R = \frac{Mem\ of\ \mathbf{H}}{Mem\ of\ (\mathbf{H^L}, \mathbf{V^H}, \Delta, \delta)} = \frac{5}{\frac{2}{\min\{B,N\}} + \frac{Q}{8} + \frac{4}{N} + \frac{1}{8}}, \tag{55}$$

*where* $\text{Minibatch} \times N$ *is the shape of activation map* $\boldsymbol{H}_l$*; $B$ denotes the block-size of LFC average pooling; and $Q$ denotes the bit-width of HFC quantization.*

*Proof.* For each mini-batch updating of normal training, a `linear`-layer caches $N$`float32` $\times$ 4byte/`float32` $= 4N$byte activation map; a Relu operator caches $N$`int8` $\times$ 1byte/`int8` $= N$byte activation map. For each mini-batch updating of DIVISION, a `linear`-layer caches $\frac{N}{\min\{B,N\}}$`bfloat16` $\times$ 2byte/`bfloat16` $= \frac{2N}{\min\{B,N\}}$byte LFC; and $QN$bit $\times \frac{1}{8}$bit/byte $= \frac{Q}{8}N$byte HFC; and spends 2`bfloat16` $\times$ 2byte/`bfloat16` $= 4$byte for $\Delta_l$ and $\delta_l$. Moreover,

a `Relu` operator caches $N$ `bit` $\times \frac{1}{8}$ `byte/bit` $= \frac{N}{8}$ `byte` activation map. Therefore, the average activation map compression rate of a `linear-Relu` block given by

$$R = \frac{4N + N}{\frac{2N}{\min\{B,N\}} + \frac{Q}{8}N + 4 + \frac{1}{8}N} = \frac{5}{\frac{2}{\min\{B,N\}} + \frac{Q}{8} + \frac{4}{N} + \frac{1}{8}}. \tag{56}$$

$\square$

## H  EXPERIMENT SETTING

We give the experiment setting including the datasets, baseline methods and model architectures in this section.

**Datasets.** We consider CIFAR-10, CIFAR-100 (Krizhevsky et al., 2009) and ImageNet (Deng et al., 2009) datasets in our experiments. **CIFAR-10:** An image dataset with 60,000 color images in 10 different classes, where each image has $32 \times 32$ pixels. **CIFAR-100:** An image dataset with 60,000 color images in 100 different classes, where each image has $32 \times 32$ pixels. **ImageNet:** A large scale image dataset which has over one million color images covering 1000 categories, where each image has $224 \times 224$ pixels.

**Baseline Methods. Normal:** Caching the exact activation map for backward propagation. **BLPA:** A systemic implementation of ACT by (Chakrabarti & Moseley, 2019), which only supports ResNet-related architectures. **AC-GC:** A framework of ACT with automatic searched bit-width for the quantization of activation maps (Evans & Aamodt, 2021). **ActNN:** Activation compression training with dynamic bit-width quantization, where the bit-allocation minimizes the variance of activation maps via dynamic processing (Chen et al., 2021). **Checkpoint:** Caching some key activation maps to reconstruct other activation maps via replaying parts of the forward pass during the backward pass (Chen et al., 2016). **SWAP:** Swapping the activation maps to the CPU during the forward pass the memory consumption of GPU, and reload the activation maps to GPU during the backward pass (Huang et al., 2020).

**DNN Architectures.** For benchmarking the model accuracy, we consider ResNet-18 (top-1 accuracy 94.89%) and ResNet-164 (top-1 accuracy 94.9%) on the CIFAR-10 dataset; DenseNet-121 (top-1 accuracy 79.75%) and ResNet-164 (top-1 accuracy 77.3%) on the CIFAR-100 dataset; and ResNet-50 (top-1 accuracy 76.15%) as well as DenseNet-161 (top-1 accuracy 77.65%) on the ImageNet dataset. Our reproduced validating accuracy is consistent with the official results of torchvision[12]. Moreover, for benchmarking the memory cost and training throughput, we consider the large models ResNet-50 and WRN-50-2 on the ImageNet dataset.

## I  IMPLEMENTATION DETAILS ABOUT DIVISION AND BASELINE METHODS

**DIVISION:** DIVISION adopts block-size 8 ($B = 8$) and 2-bit quantization ($Q = 2$) to compress the activation maps of linear, convolutional and BatchNorm layers, where the theoretical compression rate is not less than $10.35 \times$. For the operators without quantization error during backward propagation such as pooling layers, ReLu activation, and Dropout, DIVISION follows the algorithms in Appendix J to compress the activation maps. Other hyper-parameter settings are given in Table 5.

**BLPA:** Existing work (Chakrabarti & Moseley, 2019) has shown that BLPA requires at least 4-bit ACT for loss-less DNN training. We follow this setting for BLPA, where the compression rate of activation maps is not more than $8 \times$. **AC-GC:** AC-GC follows existing work (Evans & Aamodt, 2021) to take the multiplicative error $(1 + e_{\text{AC-GC}}^2) = 1.5$, where the searched bit-width enables AC-GC to satisfy this loss bound (training loss not more than $150\%$ of normal training). In this setting, AC-GC finalizes the bit-with as 7.01 after the searching, which has a nearly $4.57 \times$ compression rate of activation maps. **ActNN:** ActNN adopts 2-bit ACT and dynamic programming for searching the optimal bit-width specific for each layer, and uses per-group quantization for compressing the activation map, which has approximately $10.5 \times$ compression rate of activation maps. Such experimentally setting is denoted as L3 strategy in the original work (Chen et al., 2021), and we follow this

---

[12]https://paperswithcode.com/lib/torchvision

Table 5: Hyper-parameter setting.

| Dataset | CIFAR-10 | | CIFAR-100 | | ImageNet | |
|---|---|---|---|---|---|---|
| Architecture | ResNet-18 | ResNet-164 | ResNet-164 | DenseNet-121 | ResNet-50 | DenseNet-161 |
| Epoch | 100 | 100 | 200 | 100 | 120 | 120 |
| Batch-size | 256 | 256 | 256 | 256 | 256 | 256 |
| Initial LR | 0.1 | 0.1 | 0.15 | 0.1 | 0.1 | 0.1 |
| LR scheduler | Cos LR | Cos LR | Cos LR | Cos LR | Cos LR | Cos LR |
| Weight-decay | 0.0005 | 0.0005 | 0.0005 | 0.0005 | 0.0001 | 0.0001 |
| Optimizer | SGD | SGD | SGD | SGD | SGD | SGD |
| SGD Momentum | 0.9 | 0.9 | 0.9 | 0.9 | 0.9 | 0.9 |
| Block-size $B$ | 8 | 8 | 8 | 8 | 8 | 8 |
| Bit-width $Q$ | 2 | 2 | 2 | 2 | 2 | 2 |

```python
import torch.utils.checkpoint as checkpoint
import torch.nn as nn

class ResNet(nn.Module):
...

    def forward(self, x: Tensor) -> Tensor:
        x = self.conv1(x)
        x = self.bn1(x)
        x = self.relu(x)
        x = self.maxpool(x)
        x = checkpoint.checkpoint(self.bottleneck_layer1, x)
        x = checkpoint.checkpoint(self.bottleneck_layer2, x)
        x = checkpoint.checkpoint(self.bottleneck_layer3, x)
        x = checkpoint.checkpoint(self.bottleneck_layer4, x)
        x = self.avgpool(x)
        x = torch.flatten(x, 1)
        x = self.fc(x)
        return x

...
```

Figure 8: Implementation of checkpointed ResNet-50 and WRN-50-2.

setting in this section. **Checkpoint:** Checkpoint relies on the model-specific design of the checkpointed layers. We find that it provides a good trade-off between memory cost and running speed to checkpoint the activation map of each Botteneck block in ResNet-50 and WRN-50-2 (a ResNet-50 or WRN-50-2 has four Botteneck blocks). To implement the checkpointing of inter-media layers, we revised the forward function of the ResNet-50 and WRN-50-2 as Figure 8. **SWAP:** For the SWAP method, the memory utilization is considered both GPU and CPU because the activation maps are stored on both GPU and CPU in SWAP.

## J    COMPRESSION OF POOLING LAYERS, RELU ACTIVATIONS, AND DROPOUT

DIVISION follows Algorithms 2, 3, 4 and 5 to compress the activation map of a Max-Pooling layer, Average-Pooling layer, Relu activation and Dropout operator, respectively. For the pooling layers, we consider a simple case $\mathrm{kernelsize} = \mathrm{movingstride} = k$. General cases with different $\mathrm{kernelsize}$ and $\mathrm{movingstride}$ can be designed in analogous ways.

**Algorithm 2** Max-Pooling layer.

1: **Function** Forward ($\mathbf{H}_{l-1}$, $k$, \*\*kwargs)
2:    $\mathbf{H}_l, \mathbf{V}_{l-1}{}^{12}$ =Max-Pooling($\mathbf{H}_{l-1}$, $k$, kwargs)
3:    Pack & Cache $\mathbf{V}_{l-1}$ using Int8.
4:    **return** $\mathbf{H}_l$
5:
6: **Function** Backward($\nabla_{\mathbf{H}_l}$)
7:    Load $\mathbf{V}_{l-1}$ and $k$.
8:    $\nabla'_{\mathbf{H}_l} = \mathbf{1}_{k \times k} \otimes \nabla_{\mathbf{H}_l}$
9:    $\nabla_{\mathbf{H}_{l-1}} = \mathbf{V}_{l-1} \odot \nabla'_{\mathbf{H}_l}$
10:    **return** $\nabla_{\mathbf{H}_{l-1}}$

**Algorithm 3** Average-Pooling layer.

1: **Function** Forward ($\mathbf{H}_{l-1}$, $k$, \*\*kwargs)
2:    $\mathbf{H}_l$ =Avg-Pooling($\mathbf{H}_{l-1}$, $k$, kwargs)
3:    **return** $\mathbf{H}_l$
4:
5:
6:
7: **Function** Backward($\nabla_{\mathbf{H}_l}$)
8:    $\nabla'_{\mathbf{H}_l} = \mathbf{1}_{k \times k} \otimes \nabla_{\mathbf{H}_l}$
9:    $\nabla_{\mathbf{H}_{l-1}} = k^{-2} \nabla_{\mathbf{H}_l}$
10:    **return** $\nabla_{\mathbf{H}_{l-1}}$

**Algorithm 4** Relu operator.

1: **Function** Forward ($\mathbf{H}_{l-1}$)
2:    $\mathbf{V}_{l-1} = \text{sgn}(\mathbf{H}_{l-1})$
3:    $\mathbf{H}_l = \mathbf{V}_{l-1} \odot \mathbf{H}_{l-1}$
4:    Pack & Cache $\mathbf{V}_{l-1}$ using Int8
5:    **return** $\mathbf{H}_l$
6:
7:
8: **Function** Backward($\nabla_{\mathbf{H}_l}$)
9:    Load $\mathbf{V}_{l-1}$.
10:    $\nabla_{\mathbf{H}_{l-1}} = \mathbf{V}_{l-1} \odot \nabla_{\mathbf{H}_l}$
11:    **return** $\nabla_{\mathbf{H}_{l-1}}$

**Algorithm 5** Dropout operator.

1: **Function** Forward ($\mathbf{H}_{l-1}$)
2:    Generate a $\text{Minibatch} \times \text{Channel} \times N \times N$ binary matrix $\mathbf{V}_{l-1}$
3:    following the Bernoulli distribution with dropout probability $p$.
4:    $\mathbf{H}_l = \mathbf{V}_{l-1} \odot \mathbf{H}_{l-1}$
5:    Pack & Cache $\mathbf{V}_{l-1}$ using Int8.
6:    **return** $\mathbf{H}_l$
7:
8: **Function** Backward($\nabla_{\mathbf{H}_l}$)
9:    Load $\mathbf{V}_{l-1}$.
10:    $\nabla_{\mathbf{H}_{l-1}} = \mathbf{V}_{l-1} \odot \nabla_{\mathbf{H}_l}$
11:    **return** $\nabla_{\mathbf{H}_{l-1}}$

## K  EVALUATION OF DIVISION ON MULTI-LAYER PERCEPTRONS (MLPS)

We conduct experiments on the GAS dataset (Dua & Graff, 2017) (128-dimensional features, 13910 instances, 6 classification task). The classification model is a 4-layer MLP (128 neuros in the input layer, 6 neuros in the output layer, and 64 neuros in the hidden layer); The setting of DIVISION is $B = 16$ and $Q = 2$. The model accuracy and memory cost of activation map are given in Table 6. It is observed that DIVISION has $7.3\times$ compression rate with only $0.07\%$ degradation of model accuracy. This indicates the effectiveness of DIVISION on the MLP models.

Table 6: Model Accuracy on the GAS dataset.

| Training | Testing Accuracy (%) | Memory (KB) | Compression Rate |
|---|---|---|---|
| Normal Training | 98.92 | 250 | N/A |
| DIVISION | 98.85 | 34.2 | 7.3× |

## L  EVALUATION OF DIVISION ON VISION TRANSFORMERS

We conduct experiments to study the performance of DIVISION on Swin Transformer (Liu et al., 2021a). In this experiment, a Swin Transformer-T is trained on the ImageNet datset, where DIVISION is compared with Mesa (Pan et al., 2021), an ACT framework for visual transformer. The model accuracy and memory cost (with batch-size 128) are given in Table 7. It is observed that DIVISION achieves almost the same model accuracy compared with normal training. This indicates DIVISION can effectively compress the training of the vision Transformer.

Although DIVISION shows slightly lower accuracy lower than Mesa, the compression rate is significantly higher than Mesa. Moreover, DIVISION can be applied to general vision models, including MLPs, CNNs, and vision transformers; while Mesa is explicitly designed for vision transformers.

---

[12]$\mathbf{V}_{l-1}$ reserves the locations of each kernel-wise max-values in $\mathbf{H}_{l-1}$.

Table 7: Accuracy of Swin Transformer-T on the ImageNet dataset.

| Method | Accuracy (%) | Memory (GB) | Compression Rate |
|---|---|---|---|
| Normal Training | 81.2 | 11.81 | N/A |
| DIVISION | 81.0 | 4.19 | 2.8× |
| Mesa | 81.3 | 5.37 | 2.2× |

## M  EVALUATION OF DIVISION ON DEPTHWISE AND POINTWISE CONVOLUTIONAL LAYERS

To evaluate DIVISION on the depthwise convlution and pointwise convulution layers, we conduct experiments of MobileNet-V2 on the CIFAR-10 and CIFAR-100 datasets, where the model accuracy are given Table 8. It is observed that DIVISION achieve nearly the same accuracy compared with normal training. This indicates the effectiveness of DIVISION for the depthwise convlution and pointwise convulution layers.

Table 8: Accuracy of MoblieNet-V2 on the CIFAR-10 and CIFAR-100 datasets.

| Method | MobileNet-V2 (CIFAR-10) | MoblieNet-V2 (CIFAR-100) |
|---|---|---|
| Normal Training | 91.9 | 71.0 |
| DIVISION | 91.8 | 70.6 |

## N  RE-UTILIZATION OF HYPER-PARAMETER SETTINGS ACROSS DIFFERENT MODEL ARCHITECTURES AND DATASESTS

We conduct the follow-up experiments to study whether the hyper-parameter setting of DIVISION has a consistent effect on different model architectures and datasets. Specifically, the hyper-parameters of DIVISION are selected from $B \in \{8, 18\}$ and $Q \in \{2, 8\}$ to train ResNet-18 and MobileNet-V2 on the CIFAR-10 and CIFAR-100 datasets. The model accuracy is given in Table 9. It is observed $B$ and $Q$ have a consistent impact on different model architectures and datasets: the accuracy slightly grows with $Q$ and considerably reduces with $B$. This indicates we can reuse the hyper-parameter setting of DIVISION on CIFAR-10 to CIFAR-100 with the same model architecture, or we can reuse the setting with ResNet-18 and MobileNet-V2 on the same dataset.

Table 9: Accuracy of ResNet-18 and MoblieNet-V2 with different hyper-parameter settings.

| | $B = 18\ Q = 2$ | $B = 8\ Q = 2$ | $B = 8\ Q = 8$ |
|---|---|---|---|
| ResNet-18 CIFAR-10 | 78.7 | 94.6 | 94.9 |
| ResNet-18 CIFAR-100 | 73.2 | 76.9 | 77.0 |
| MobileNet-V2 CIFAR-10 | 10.0 | 91.7 | 91.0 |
| MobileNet-V2 CIFAR-100 | 62.4 | 70.6 | 71.6 |

## O  COMPARISON OF DIVISION WITH CHECKPOINT STRATEGY OF MEGATRON-LM

DIVISION is compared with the checkpointing strategy of Megatron-LM (Shoeybi et al., 2019). According to the official guideline, Megatron-LM checkpoints the activation map after each transformer block. We follow this strategy to checkpoint the activation map after each transformer block in the Swin Transformer, and after each Bottleneck block in the ResNet-50. The memory cost (on the ImageNet dataset with batch-size 128) is given in Table 10. It is observed DIVISION has a higher compression rate (2.8× for Swin Transformer-T and 7.9× for ResNet-50) than the Check-point strategy of Megatron-LM (2.3× for Swin Transformer-T and 2.27× for ResNet-50). This indicates the effectiveness of DIVISION over the checkpoint strategy of Megatron-LM.

Table 10: Memory cost of DIVISION and Checkpoint strategy of Megatron-LM.

| Memory cost (GB) | Normal Training | DIVISION | Checkpoint strategy of Megatron-LM |
|---|---|---|---|
| Swin Transformer-T | 11.81 | 4.19 (2.8×) | 5.14 (2.3×) |
| ResNet-50 | 10.62 | 1.35 (7.9×) | 4.68 (2.27×) |

## P    EFFECT OF BIT-WIDTH ON THE FIXED QUANTIZATION

To demonstrate the effectiveness of dual activation precision, DIVISION ($B = 8$, $Q = 2$) is compared with the fixed quantization under different bit-width. The model accuracy of ResNet-50 on the ImageNet dataset is given in Table 11. It is observed the training fails to converge to an optimal solution under 2-bit quantization, even though it performs favorably under 4-bit and 8-bit quantization. This result is consistent with existing work (Chen et al., 2021) (Table 3 in (Chen et al., 2021)). In contrast, DIVISION achieves 75.9% accuracy when adopting 2-bit quantization to compress the HFC. This indicates the effectiveness of dual activation precision in terms of the model accuracy under low bit-width quantization.

Table 11: Model accuracy of fixed quantization under different bit-width.

| Bit-width | 2 | 4 | 8 |
|---|---|---|---|
| Fixed Quantization | 0.1 | 76.05 | 76.35 |
| DIVISION | 75.9 | omit | omit |

## Q    EFFECT OF MINI-BATCH SIZE ON THE THROUGHPUT

As a supplementary of Section 5.5, a follow-up experiment is conducted to study the effect of batch-size on the training throughput. The result on the CIFAR-10 dataset is given in Table 12. It is observed: for the experiment w/o AMP, the throughput significantly grows as the batch-size grows from 64 to 128, but is almost unchanged when the batch-size $\geq$ 128; for the experiment w/ AMP, it grows continuously when the batch-size grows from 64 to 256, and becomes stable when the batch-size $\geq$ 256.

Table 12: Traininig throughput versus the batch-size.

| images/s | Batchsize=64 | Batchsize=128 | Batchsize=256 | Batchsize=512 | Batchsize=1024 |
|---|---|---|---|---|---|
| w/o AMP | 1185.27 | 2184.36 | 2335.41 | 2394.84 | 2409.99 |
| w/ AMP | 1273.46 | 2285.87 | 3753.06 | 3981.95 | 4068.67 |

Intuitively, as the batch-size grows, the GPU can parallel process more images per second, until the GPU is 100% utilized (Goyal et al., 2017). In the experiment w/o AMP, the GPU is almost 100% utilized when batch-size $\geq$ 128; while this happens when batch-size $\geq$ 256 in the experiment w/ AMP. More images can be processed in the experiment w/ AMP, since it employs bfloat16 operations in the training, in contrast with the float32 operations in the training w/o AMP, where a bfloat16 operation has nearly half of the computation cost of a float32 operation. Therefore, the training throughput significantly grows from 2086 (images/s) to 3753 (images/s) as the batch-size grows from 128 to 256 in the experiment w/ AMP, but is almost unchanged (2184 images/s vs 2335 images/s) in the same condition in the experiment w/o AMP.

## R    DETAILS ABOUT THE COMPUTATION INFRASTRUCTURE

The details about our physical computing infrastructure for testing the training memory cost and throughput are given in Table 13.

Table 13: Computing infrastructure for the experiments.

| Device Attribute | Value |
|---|---|
| Computing infrastructure | GPU |
| GPU model | Nvidia-RTX3090 |
| GPU number | 1 |
| GPU Memory | 24567 MB |

## S  IMPLEMENTATION DETAILS OF THE EXPERIMENT IN SECTION 5.4

We give the implementation details of the experiment in Section 5.4. Specifically, *DIVISION w/o HFC* takes block-size $B=4$ for estimating LFC; *DIVISION w/o LFC* takes the bit-width $Q=2$ for the quantization of HFC; DIVISION combines these settings for the training; and *Fixed Quant* has a 2-bit per-group quantization of activation maps during the backward pass of the training, where the group size of quantization is 256. Other training hyper-parameters are given in Table 5.

