# OpenReview forum: "DIVISION: Memory Efficient Training via Dual Activation Precision"
_ICLR.cc/2023/Conference — Submitted to ICLR 2023_

### Official Review · Reviewer_Eerz · 2022-10-23

**Confidence:** 5
**Correctness:** 2
**Technical Novelty And Significance:** 3
**Empirical Novelty And Significance:** 1
**Recommendation:** 3

**Clarity, Quality, Novelty And Reproducibility:**

The proposed method is well-motivated and has a clear insight. However, the experiments and evaluations are not convincing, especially the comparisons with baselines. The improvement over baseline is also minor. Overall, this paper needs significant revision in order to make the experimental results more clear.

**Strength And Weaknesses:**

### Strength

1. This paper has a clear insight and motivation. In the experiments, the authors have shown that low-frequency components are more important for gradient computation than high-frequency components. To this end, the proposed DIVISION is well-motivated. The idea is also novel in the literature.
2. The ablation studies are comprehensive. For example, Section 5.5 has clearly shown the effect of different hyperparameters in the proposed model.
3. This paper is well written and easy to follow.

### Weakness

Overall, I appreciate the idea and novelty of this paper. However, the experiments and evaluations in this paper need major revision. Some results and observations are quite unusual. For example,

1. I have experimented with ActNN (using official code) and Checkpointing (on both Vision Transformers and CNNs) to check the correctness of the experiment. In my observation, checkpointing is a strong baseline which should be faster than many ACT frameworks. However, the large gap between ActNN and Checkpointing in Figure 6 (a) is confusing. Moreover, according to the provided experimental settings (Figure 8), I compared checkpointing with ActNN (L3) based on ResNet-50 on my local RTX 3090 machine.  With a batch size of 64 and 224 $\times$ 224 images, the results show that checkpointing is faster (310 images/s v.s. 280 images/s) than ActNN under a similar amount of memory-saving (3.2GB v.s. 3.4GB). Therefore, I keep skeptical about Figures 6 (a) and (b).
2. In Figure 6 (c), fixed quantization achieves much worse performance, which is very unnatural. Fixed-bit quantization in ActNN performs favorably on ResNet. For example, L2 of ActNN quantizes activation into fixed 4 bits, but it still performs well. The authors need to give a detailed explanation here.
3. In Figure 7 (b), it is counter-intuitive that using a batch size of 128 significantly slows down the throughput compared with using a batch size of 256. This figure also needs to be explained further.
4. Despite the ablation studies, the main experiment in Table 1 shows that the memory-saving gain compared with ActNN is very minor.
5. Apart from CNNs, how is the performance of DIVISION on Vision Transformers? And how do you compare with GACT [A] and Mesa [B]?


### References
[A] Liu, Xiaoxuan, et al. "GACT: Activation compressed training for generic network architectures.", *International Conference on Machine Learning (ICML) 2022*.

[B] Pan, Zizheng, et al. "Mesa: A Memory-saving Training Framework for Transformers." *arXiv preprint arXiv:2111.11124* (2021).


**Summary Of The Paper:**

This paper introduced an activation compressed training (ACT) framework, called DIVISION. Specifically, DIVISION is inspired by the insight that DNN backward propagation mainly utilizes the low-frequency component (LFC) of the activation maps, while the majority of memory is for the storage of the high-frequency component (HFC) during the training. In this case, the proposed method stores two memory-efficient copies of activations instead of the original activations, which are average-pooled low-frequency activations and low-bit quantized high-frequency signals. At the time of backpropagation, DIVISION recovers the original size of activations by upsampling low-frequency signals and dequantizing high-frequency signals. Extensive experiments have shown the proposed method works and achieves faster training throughput than related baselines


**Summary Of The Review:**

Some experimental results are clearly not convincing, which leads to rejection.

---

> ### Author Response · Authors · 2022-11-13
> **Response to Reviewer Eerz [Part 1/3 Q1]**
>
> We thank the reviewer for the thoughtful suggestions and detailed reviews.
>
> **[Q1] I have experimented with ActNN (using official code) and Checkpointing (on both Vision Transformers and CNNs) to check the correctness of the experiment. In my observation, checkpointing is a strong baseline which should be faster than many ACT frameworks. However, the large gap between ActNN and Checkpointing in Figure 6 (a) is confusing. Moreover, according to the provided experimental settings (Figure 8), I compared checkpointing with ActNN (L3) based on ResNet-50 on my local RTX 3090 machine. With a batch size of 64 and 224  224 images, the results show that checkpointing is faster (310 images/s v.s. 280 images/s) than ActNN under a similar amount of memory-saving (3.2GB v.s. 3.4GB). Therefore, I keep skeptical about Figures 6 (a) and (b).**
>
>
> We appreciate the reviewer for this instructive comment. We double-check the configuration of the Checkpoint experiment, and update the environment such that all methods are tested in the same condition. After this, we re-conduct the throughput experiment of all methods with ResNet-50 and WRN-50-2 (on a single RTX 3090 GPU), and give the results in the following two tables. It is observed Checkpoint is indeed faster than ActNN in the training of ResNet-50, as mentioned by the reviewer; however, ActNN is slightly faster than Checkpoint in the training of WRN-50-2. This is because WRN-50-2 has a more time-consuming forward pass than ResNet-50, which makes the recomputation of activation maps in Checkpoint not that fast.
>
> We have updated Figures 6 (a) and (b), and the discussions in Section 5.3 accordingly. Moreover, we also open-source the code of the Checkpoint experiment (https://anonymous.4open.science/r/Checkpoint_ImageNet-EFCC/README.md). We sincerely hope this can address the concern of the reviewer.
>
> | ResNet-50 |  Batchsize=64 | Batchsize=128 | Batchsize=256 | Batchsize=512 |
> | :----: | :----: | :----: | :----: | :----: |
> | Checkpoint | $334.4$ | $346.0$ | $352.5$ | $354.9$ |
> | ActNN | $266.2$ | $296.3$ | $308.3$ | $315.6$ |
> | DIVISION | $369.8$ | $378.9$ | $380.7$ | $380.8$ |
> | Normal | $488.6$ | $511.9$ | $516.1$ | OOM |
>
> | WRN-50-2 |  Batchsize=64 | Batchsize=128 | Batchsize=256 | Batchsize=512 |
> | :----: | :----: | :----: | :----: | :----: |
> | Checkpoint | $187.5$ | $196.7$ | $197.3$ | $199.8$ |
> | ActNN | $191.4$ | $206.5$ | $212.3$ | $215.9$ |
> | DIVISION | $239.1$ | $250.8$ | $252.7$ | $253.4$ |
> | Normal | $307.7$ | $325.1$ | $328.8$ | OOM |

---

> ### Author Response · Authors · 2022-11-13
> **Response to Review Eerz [Part 2/3 Q2]**
>
> **[Q2.1] In Figure 6 (c), fixed quantization achieves much worse performance, which is very unnatural. Fixed-bit quantization in ActNN performs favorably on ResNet. For example, L2 of ActNN quantizes activation into fixed 4 bits, but it still performs well. The authors need to give a detailed explanation here.**
>
> **[Q2.2] In Figure 7 (b), it is counter-intuitive that using a batch size of 128 significantly slows down the throughput compared with using a batch size of 256. This figure also needs to be explained further.**
>
> **[Answer to Q2.1]**
>
> We thank the reviewer for this thoughtful comment. To give a detailed explanation, we conduct experiments of fixed quantization under different bit-width (using ResNet-50 on the ImageNet dataset). The model accuracy is given in the following table.
>
> | | 2bit | 4bit | 8bit |
> | :----------------: | :----------------: | :----------------: | :----------------: |
> Accuracy (%) | $0.1$ | $76.05$ | $76.35$ |
>
> It is observed the training fails to converge to an optimal solution under 2-bit quantization, even though it performs favorably under 4-bit and 8-bit quantization. We believe this result is correct because it is consistent with existing work [1] (Table 3 in [1]).
> Since our DIVISION adopts 2-bit quantization for compressing the HFC of activation maps, we only compare it with 2-bit fixed quantization in Figure 6 (c) such that they have nearly the same memory cost. The experiment settings are highlighted in Appendix S.
> We sincerely hope this result can address the concern of the reviewer.
>
> **[Answer to Q2.2]**
>
> We thank the reviewer for this thoughtful comment. To give a further explanation, we conduct experiments to analyze the throughput versus different training batch-size. The result is given in the following table.
> It is observed: for the experiment w/o AMP (automatic mixed precision [2]), the throughput significantly grows as the batch-size grows from 64 to 128, but is almost unchanged when the batch-size >= 128;
> for the experiment w/ AMP, it grows continuously when the batch-size grows from 64 to 256, and becomes stable when the batch-size >= 256.
>
> | images/s | Batchsize=64 | Batchsize=128 | Batchsize=256 | Batchsize=512 | Batchsize=1024 |
> | :---: | :---: | :---: | :---: | :---: | :---: |
> | w/o AMP | $1185.27$ | $2184.36$ | $2335.41$ | $2394.84$ | $2409.99$
> | w/ AMP | $1273.46$ | $2285.87$ | $3753.06$ | $3981.95$ | $4068.67$
>
> Intuitively, as the batch-size grows, the GPU can parallel process more images per second, until the GPU is 100% utilized [3].
> In the experiment w/o AMP, the GPU is almost 100% utilized when batch-size >= 128; while this happens when batch-size >= 256 in the experiment w/ AMP.
> More images can be processed in the experiment w/ AMP, since it employs bfloat16 operations in the training, in contrast with the float32 operations in the training w/o AMP, where a bfloat16 operation has nearly half of the computation cost of a float32 operation.
> Therefore, the training throughput significantly grows from 2086 (images/s) to 3753 (images/s) as the batch-size grows from 128 to 256 in the experiment w/ AMP, but is almost unchanged (2184 images/s vs 2335 images/s) in the same condition in the experiment w/o AMP.
> We sincerely hope the analysis can address the concern of the reviewer.
>
> [1] Jianfei Chen et al. ActNN: Reducing Training Memory Footprint via 2-Bit Activation Compressed Training
>
> [2] Sharan Narang et al. Mixed Precision Training
>
> [3] Priya Goyal et al. Accurate, Large Minibatch SGD: Training ImageNet in 1 Hour

---

> ### Author Response · Authors · 2022-11-14
> **Response to Review Eerz [Part 3/3 Q3 and Q4]**
>
> **[Q3] The main experiment in Table 1 shows that the memory-saving gain compared with ActNN is very minor.**
>
> We thank the reviewer for this comment. We agree with the reviewer that DIVISION has limited memory-saving gain compared with ActNN. However, we would like to highlight the contribution of DIVISION is a **simpler** and **more transparent** framework than existing work (e.g. ActNN, AC-GC, etc.). We believe **improving the simplicity and transparency of ML algorithms is important for both the community and follow-up research**. Compared with ActNN, DIVISION compresses the activation map using average pooling and fixed bit-with per-channel quantization. In contrast, ActNN searches for the optimal bit-width to match different training samples, and applies per-group quantization based on the searched bit-width. The simplicity of DIVISION enables efficient compression and decompression (see Figures 6 (a) and (b)); and the transparency allows us to understand the compressible (HFC) and non-compressible factors (LFC) during DNN training. From this perspective, we believe our work has non-trivial contributions to the ML community.
>
> **[Q4] Apart from CNNs, how is the performance of DIVISION on Vision Transformers? And how do you compare with GACT and Mesa?**
>
> We thank the reviewer for this constructive comment.
> To show the performance of DIVISION on the vision transformers, we conduct experiments of the Swin Transformer [1] on the ImageNet dataset.
> Specifically, we adopt normal training, DIVISION, and Mesa to train the Swin Transformer-T on the ImageNet dataset, and give the model accuracy and memory cost (batch-size 128) in the following table. It is observed that DIVISION achieves almost the same model accuracy compared with normal training. This indicates the effectiveness of DIVISION on the vision transformer.
>
> | Method | Accuracy (\%) | Memory (GB) | Compression Rate
> | :----: | :----: | :----: | :-----: |
> | Normal Training | $81.2$ | $11.81$ | N/A |
> | DIVISION | $81.0$ | $4.19$ | $2.8\times$ |
> | Mesa | $81.3$ | $5.37$ | $2.2\times$ |
>
>
> Although DIVISION shows slightly lower accuracy lower than Mesa [2], the compression rate is higher than Mesa. Moreover, DIVISION can be applied to general vision models, including MLPs, CNNs, and vision transformers; while Mesa is explicitly designed for vision transformers.
>
> [1] Ze Liu et al. Swin Transformer: Hierarchical Vision Transformer using Shifted Windows
>
> [2] Zizheng Pan et al. A Memory-saving Training Framework for Transformers.

---

> ### Author Response · Authors · 2022-11-18
> **To Reviewer Eerz**
>
> We thank the reviewer for the instructive comments on our work. We spent quite amount of time and effort in collecting more experiment results, trying our best to address the reviewer's concerns.
> As the deadline of context revising (Nov. 18th) is coming, would you please let us know if there are more questions/concerns left? It will be our pleasure to receive your follow-up feedback.

---

> ### Author Response · Authors · 2022-11-29
> **To Reviewer Eerz: Are there more questions/concerns to Answer?**
>
> We thank the reviewer for the efforts in the paper review. We sincerely value your comments, and provide more explanations and experiment results to address your concerns. It has been two weeks since we posted our response. Would you please let us know if you have more questions/concerns about this work? It will be our pleasure to answer your follow-up question.

---

> ### Author Response · Authors · 2022-12-07
> **To Reviewer Eerz: Are there more questions/concerns left?**
>
> We thank the reviewer for the efforts in the paper review. We sincerely value your comments, and provide more explanations and experiment results to address your concerns. As the deadline of the discussion stage is coming, Would you please let us know if there are more questions/concerns left? It will be our pleasure to answer your follow-up question.

---

> ### Author Response · Authors · 2022-12-09
> **Last message to Reviewer Eerz: Are there more questions/concerns left?**
>
> We thank the reviewer for the efforts in the paper review, and we have provided detailed explanations and experiment results to address your concerns. As the discussion stage will be ended in three days, would you please let us know if there are more questions/concerns left? We will believe no concern is left if you keep no more feedback on our response until the end of the discussion.

---

> > ### Comment · Reviewer_Eerz · 2022-12-12
> > **Response to the feedback**
> >
> > Thank the authors for the additional analysis and experiment results. I have carefully read the response as well as the comments from the other reviewers.
> >
> > I appreciate the overall idea of DIVISION. It is clear and well-motivated. However, my main concern is still the minor memory-saving gain compared to previous works. The improvement of the speed-memory trade-off for activation compressed training (ACT) frameworks is also not significant. I agree with the authors that ML transparency is important for the community. However, in my opinion, the basic idea of ACT frameworks is already quite clear, i.e., compressing the saved activations into low-bit values for memory-saving.
> >
> > For Q1, it seems the previous evaluation for ActNN is indeed incorrect. Besides, according to the further explanation, I assume the advantage of DIVISION may also only show in some specific architectures that have very complicated computational graphs, limiting its versatility.
> >
> > For Q4, the authors do not address my concern about comparing DIVISION with GACT.
> >
> > Considering the above, I tend to keep my initial rating.

---

> > > ### Author Response · Authors · 2022-12-13
> > > **To reviewer Eerz**
> > >
> > >
> > > We thank the reviewer for the feedback. We sincerely value your comments and provide follow-up explanations to address your concerns [[here]](https://openreview.net/forum?id=6Pv8AMSylux&noteId=4voQ1KCs0td).
> > > We sincerely hope our response can address your concerns.
> > > It will be our pleasure to receive your follow-up feedback.

---

### Official Review · Reviewer_PUeD · 2022-10-24

**Confidence:** 4
**Correctness:** 3
**Technical Novelty And Significance:** 3
**Empirical Novelty And Significance:** 3
**Recommendation:** 6

**Clarity, Quality, Novelty And Reproducibility:**

Clarity: the paper is clearly written, with a detailed supplementary material.
Quality: the technic sounds, though the contribution might be somewhat limited.
Novelty: applying DCT for ACT is somewhat novel. However, as DCT is also applied frequently for computer vision previously (such as JPEG-ACT). The novelty might not be too much.
Reproducibility: the reproducibility is good.

**Strength And Weaknesses:**

Strength:
- Adopting frequency-domain features for ACT is somewhat novel.
- The proposed approach is simple and effective.
- The presentation is clear.

Weaknesses:
- The proposed approach might be limited to convolutional networks, or at least assuming that the feature map has the shape [N, C, H, W].

The authors can argue that the proposed method can be generalized to 1D or 3D problems with 1D/3D pooling. But as the feature statistics can differ across problems with different dimensions, whether the proposed approach is also effective for 1D/3D problems is still unclear. Moreover, I am not sure if the proposed approach can be applied to MLPs (just pure MLP, not MLP-Mixer, which are in fact 1x1 convolution), as MLPs are permutation-invariant.

To make the paper stronger, the authors might want to test on more transformer architectures, including the vision ones (Swin, etc.) and the text ones (such as BERT).

- The significance might be still somewhat limited. There are some improvements over ActNN, like the accuracy of DenseNet-161 and the throughput. But I am not sure if the improvement is significant enough for an ICLR publication. The authors might also want to compare with more recent checkpointing approaches, such as Mesa and the hand-crafted checkpointing strategy for transformers (like those in Megatron-LM and DeepSpeed).

Other questions:
- Does the proof of Theorem 1 has anything to do with DCT or the frequency domain? I suspect that simply showing that |H^L - H| < |H^H - H| might be also sufficient to prove Theorem 1?

- Should the window size be different for feature maps of different resolution? For example, the final 8x8 feature maps might require a smaller window size.

**Summary Of The Paper:**

This paper proposes a method for activation compressed training. The proposed method, DIVISION, takes a frequency-domain view of the activation map. They pointed out that high-frequency features are more robust to quantization than the low-frequency ones. To utilize this observation, the authors extracts low-frequency features (approximately) by average pooling, and only quantizes the high-frequency residual. They present a theorem to show only perseving low-frequency features yields tighter gradient error bound that only perserving high-frequency ones. The experiments are done for ResNets and DenseNets. Compared to the previous approach ActNN, the proposed method can achieve better accuracy and higher throughput due to its simplicity (does not need to estimate the per-tensor precision).

**Summary Of The Review:**

The paper does make improvements to ACT for deep learning. The main weaknesses are the applicable range of the proposed method and the significance.

---

> ### Author Response · Authors · 2022-11-13
> **Response to Reviewer PUeD [Part 1/4 Q1]**
>
>
> We thank the reviewer for the detailed comments.
>
> **[Q1.1] The proposed approach might be limited to convolutional networks, or at least assuming that the feature map has the shape $[N, C, H, W]$.**
>
> **[Q1.2] The authors can argue that the proposed method can be generalized to 1D or 3D problems with 1D/3D pooling. But as the feature statistics can differ across problems with different dimensions, whether the proposed approach is also effective for 1D/3D problems is still unclear.**
>
> **[Q1.3] Moreover, I am not sure if the proposed approach can be applied to MLPs (just pure MLP, not MLP-Mixer, which are in fact 1x1 convolution), as MLPs are permutation-invariant.**
>
>
> **[Answer to Q1.1]**
>
> We thank the reviewer for expressing this concern. Although our analysis in Sections 4.1 and 4.2 assumes the activation map is in the shape of $[N, C, H, W]$, our framework can be easily extended to 1D/3D cases. As a supplementary, we give the computational details of DIVISION in processing 1D/3D activation maps in Appendix F. Moreover, we opensource the code of DIVISION for MLP (https://anonymous.4open.science/r/division-MLP-9416/README.md) and visual transformer (https://anonymous.4open.science/r/division-swin-transformer-F83E/README.md). Finally, we have more experiment results to demonstrate the effectiveness of DIVISION on the visual transformer and MLPs (which have 1D/3D activation maps) in the answer to [Q1.2] and [Q1.3].
>
> **[Answer to Q1.2]**
>
> We agree with the reviewer that 1D/3D activation maps may have different feature statistics with 2D activation maps.
> To study whether DIVISION is effective for 1D/3D activation maps, we conduct  experiments of training the Swin Transformer [1] (Swin Transformer-T) on the ImageNet dataset. The model accuracy is given in the following table. It is observed that DIVISION achieves the same level of model accuracy compared with normal training. This indicates DIVISION is also effective for the models with 1D/3D activation maps in terms of the model accuracy and compression rate.
>
> | Method | Accuracy (\%) | Memory (GB) | Compression Rate
> | :----: | :----: | :----: | :-----: |
> | Normal Training | $81.2$ | $11.81$ | N/A |
> | DIVISION | $81.0$ | $4.19$ | $2.8\times$ |
>
>
> **[Answer to Q1.3]**
>
> We would like to clarify that DIVISION is also effective for pure MLPs.
> To study the performance of DIVISION on pure MLPs, we conduct experiments on the GAS dataset [2] (a tabular dataset, with 128-dimensional features, 13910 instances, and 6-class labels).
> The classification model is a 4-layer MLP (128 neurons in the input layer, 6 neurons in the output layer, and 64 neurons in the hidden layer); The setting of
> DIVISION is $B=16$ and $Q=2$. The model accuracy and memory cost (batch-size 256) of activation map is given in the following table. It is observed that DIVISION has a $7.3\times$ compression rate with only $0.07\%$ degradation of model accuracy. This indicates the effectiveness of DIVISION on the MLP models.
> (Although it may not require compression to train such a tiny model, without a doubt, the experiment results show our proposed DIVISION can provide almost loss-less training for pure MLP models.)
>
> | Training | Testing Accuracy (\%) | Memory (KB) | Compression Rate
> | :-----: | :-----: | :-----: | :-----: |
> | Normal Training | $98.92$  |  $250$ | N/A |
> | DIVISION | $98.85$ |  $34.2$ | $7.3\times$
>
> [1] Ze Liu et al. Swin Transformer: Hierarchical Vision Transformer using Shifted Windows
>
> [2] UCI dataset: https://archive.ics.uci.edu/ml/datasets/Gas+Sensor+Array+Drift+Dataset+at+Different+Concentrations

---

> ### Author Response · Authors · 2022-11-13
> **Response to Review PUeD [Part 2/4 Q2]**
>
> **[Q2.1] The significance might be still somewhat limited. There are some improvements over ActNN, like the accuracy of DenseNet-161 and the throughput. But I am not sure if the improvement is significant enough for an ICLR publication.**
>
> **[Q2.2] The authors might also want to compare with more recent checkpointing approaches, such as Mesa and the hand-crafted checkpointing strategy for transformers (like those in Megatron-LM and DeepSpeed).**
>
> **[Answer to Q2.1]**
>
> We thank the reviewer for this comment. We would like to highlight the contribution of DIVISION is a **more simple** and **transparent** framework than existing work (e.g. ActNN, AC-GC, etc.). We believe **improving the simplicity and transparency of ML algorithms is important for both the community and follow-up research**. Taking the comparison with ActNN as an example, DIVISION compresses the activation map using average pooling and fixed bit-width per-channel quantization. In contrast, ActNN relies on searching optimal bit-width to match different training samples, and per-group quantization based on the searched bit-width. The simplicity of DIVISION enables efficient compression and decompression (see Figures 6 (a) and (b)); and transparency allows us to understand the compressible (HFC) and non-compressible factors (LFC) during DNN training. From this perspective, we believe our work has a non-trivial contribution to the ML community.
>
> **[Answer to Q2.2]**
>
> We thank the reviewer for this thoughtful comment. We believe the reviewer has a misunderstanding about the existing work Mesa [1]. **Mesa is an ACT-based memory-efficient training method** explicitly designed for vision transformers, rather than a checkpointing method. While DIVISION can be applied to general vision models, including MLPs, CNNs, and vision transformers.
>
> To compare the performance of DIVISION with Mesa on the vision transformer, we conduct the experiments of training the Swin Transformer [2] (Swin Transformer-T) on the ImageNet dataset. The model accuracy and memory cost (with batch-size 128) is given in the following table. It is observed that DIVISION achieves almost the same accuracy compared with normal training and Mesa. Moreover, DIVISION has a higher compression rate ($2.8\times$) than Mesa ($2.2\times$). This indicates DIVISION can effectively compress the training of the vision transformer.
>
> | Method | Accuracy (\%) | Memory (GB) | Compression Rate
> | :----: | :----: | :----: | :-----: |
> | Normal Training | $81.2$ | $11.81$ | N/A |
> | DIVISION | $81.0$ | $4.19$ | $2.8\times$ |
> | Mesa | $81.3$ | $5.37$ | $2.2\times$ |
>
> Moreover, we compare DIVISION with the checkpoint strategy of Megatron-LM [3]. According to the official guideline, Megatron-LM checkpoints the activation map after each transformer block. We follow this strategy to checkpoint the activation map after each transformer block in the Swin Transformer, and after each Bottleneck block in the ResNet-50. The memory cost (on the ImageNet dataset with batch-size 128) is given in the following table. It is observed DIVISION has a higher compression rate ($2.8\times$ for Swin Transformer-T and $7.9\times$ for ResNet-50) than the checkpoint strategy of Megatron-LM ($2.3\times$ for Swin Transformer-T and $2.27 \times$ for ResNet-50). This indicates the effectiveness of DIVISION over the checkpoint strategy of Megatron-LM.
>
> | Memory cost (GB) | Normal Training | &emsp; DIVISION &emsp; | Checkpoint strategy of Megatron-LM [3] |
> | :----: | :----: | :----: | :-----: |
> | Swin Transformer-T | $11.81$ | $4.19~(2.8\times)$ | $5.14~(2.3\times)$ |
> | ResNet-50 | $10.62$ | $1.35~(7.9\times)$ | $4.68~(2.27 \times)$
>
> We are afraid that DeepSpeed has not open-sourced its checkpointing strategy in its publications [4]. We sincerely hope our comparison with Mesa and Megatron-LM can address the concern of the reviewer.
>
> [1] Zizheng Pan et al. A Memory-saving Training Framework for Transformers
>
> [2] Ze Liu et al. Swin Transformer: Hierarchical Vision Transformer using Shifted Windows
>
> [3] Mohammad Shoeybi et al. Megatron-LM: Training multi-billion parameter language models using model parallelism
>
> [4] Jeff Rasley et al. DeepSpeed: System Optimizations Enable Training Deep Learning Models with Over 100 Billion Parameters

---

> ### Author Response · Authors · 2022-11-13
> **Response to Review PUeD [Part 3/4 Q3]**
>
> **[Q3.2] I suspect that simply showing that $||{\bf{H}}^L - {\bf{H}}||_F < ||{\bf{H}}^H - {\bf{H}}||_F$ might be also sufficient to prove Theorem 1?**
>
> **[Answer to Q3.1]**
>
> We thank the reviewer for raising this good question. We would like to clarify it is necessary to adopt DCT in Theorem 1.
> Although $||{\bf{H}}^L - {\bf{H}}||_F < ||{\bf{H}}^H - {\bf{H}}||_F$ is the same as $||\tilde{\bf{H}}^L - \tilde{\bf{H}}||_F < ||\tilde{\bf{H}}^H - \tilde{\bf{H}}||_F$, it is difficult to sparsify ${\bf{H}}^L$ in the spatial domain.
> Note that our goal is to compress ${\bf{H}}^L$ during DNN training, ${\bf{H}}^L$ should be a sparse matrix for saving the memory.
> Fortunately, the DCT provides an ideal solution given by $\tilde{\bf{H}} = \rm{DCT}(\bf{H})$ and $\tilde{\bf{H}}^L = \tilde{\bf{H}} \odot \bf{M}$, where $\mathbf{M}$ denotes a loss-pass mask with very few elements 1 in the upper-left corner. In this way, $\tilde{\bf{H}}^L$ can not only satisfy $||\tilde{\bf{H}}^L - \tilde{\bf{H}}||_F < ||\tilde{\bf{H}}^H - \tilde{\bf{H}}||_F$ during DNN training, but also easy to compress ($\tilde{\bf{H}}^L$ is a sparse matrix).
>
> **[Answer to Q3.2]**
>
> We thank the reviewer for this comment. We would like to clarify $||{\bf{H}}^L - {\bf{H}}||_F < ||{\bf{H}}^H - {\bf{H}}||_F$ cannot prove Theorem 1. We believe the reviewer has a misunderstanding about Section 3.2. The objective of Section 3.2 is to demonstrate a DNN satisfies $\rm{GEB}^L < \rm{GEB}^H$ during the training.
> To achieve this goal, the experiment provides empirical results $||\tilde{\bf{H}}^L - \tilde{\bf{H}}||_F < ||\tilde{\bf{H}}^H - \tilde{\bf{H}}||_F$ can be satisfied during DNN training; and Theorem 1 provides a theoretical derivation from $||\tilde{\bf{H}}^L - \tilde{\bf{H}}||_F < ||\tilde{\bf{H}}^H - \tilde{\bf{H}}||_F$ to $\rm{GEB}^L < \rm{GEB}^H$.
> Therefore, both $||\tilde{\bf{H}}^L - \tilde{\bf{H}}||_F < ||\tilde{\bf{H}}^H - \tilde{\bf{H}}||_F$ and Theorem 1 are necessary for arriving at the conclusion.
> It might be confusing that the reviewer claims $||{\bf{H}}^L - {\bf{H}}||_F < ||{\bf{H}}^H - {\bf{H}}||_F$ can simply prove Theorem 1.
>
>
> Beyond the empirical results in Section 3.2, we provide more experiment results to demonstrate that it satisfies $||\tilde{\bf{H}}^L - \tilde{\bf{H}}||_F < ||\tilde{\bf{H}}^H - \tilde{\bf{H}}||_F$ during DNN training. Specifically, the value of $||\tilde{\bf{H}}^L - \tilde{\bf{H}}||_F \big/ ||\tilde{\bf{H}}^H - \tilde{\bf{H}}||_F$ is estimated during the training (epochs 20, 40, and 60) of ResNet-18 and DenseNet-121; $\bf{H}$ takes the input activation maps of the four BasicBlocks in ResNet-18 and four DenseBlocks in DenseNet-121; $\tilde{\bf{H}} = \rm{DCT}(\bf{H})$; $\tilde{\bf{H}}^L = \tilde{\bf{H}} \odot \bf{M}$; and only $4\%$ of the elements in the upper-left corner of $\mathbf{M}$ is $1$, such that $\tilde{\bf{H}}^L$ is sparse enough. The results of ResNet-18 and DenseNet-121 are given in the first and second table, respecitively.
> It is observed that $||\tilde{\bf{H}}^H - \tilde{\bf{H}}||_F \big/ ||\tilde{\bf{H}}^L - \tilde{\bf{H}}||_F < 1$ consistently holds for different layers of ResNet-18 and DenseNet-121 in different training epochs.
> Based on this experiment result $||\tilde{\bf{H}}^L - \tilde{\bf{H}}||_F < ||\tilde{\bf{H}}^H - \tilde{\bf{H}}||_F$, we can furtherly use Theorem 1 to achieve $\rm{GEB}^L < \rm{GEB}^H$ during DNN training.
>
>
> | Epoch | $20$ | $40$ | $60$ | Average |
> | :-----: | :-----: | :-----: | :-----: | :-----: |
> | BasicBlock-1 | $0.185$  | $0.169$ | $0.163$ | $0.172$ |
> | BasicBlock-2 | $0.649$  | $0.620$ | $0.587$ | $0.619$ |
> | BasicBlock-3 | $0.681$  | $0.725$ | $0.692$ | $0.699$ |
> | BasicBlock-4 | $0.463$  | $0.528$ | $0.505$ | $0.499$ |
> | Average | $0.495$ | $0.511$ | $0.487$ | $0.498$ |
>
>
> | Epoch | $20$ | $40$ | $60$ | Average |
> | :-----: | :-----: | :-----: | :-----: | :-----: |
> | DenseBlock-1 | $0.425$ | $0.432$ | $0.425$ | $0.427$ |
> | DenseBlock-2 | $0.365$ | $0.370$ | $0.368$ | $0.368$ |
> | DenseBlock-3 | $0.454$ | $0.372$ | $0.387$ | $0.404$ |
> | DenseBlock-4 | $0.142$ | $0.135$ | $0.122$ | $0.133$ |
> | Average | $0.347$ | $0.327$ | $0.326$ | $0.333$ |

---

> ### Author Response · Authors · 2022-11-13
> **Response to Review PUeD [Part 4/4 Q4]**
>
> **[Q4] Should the window size be different for feature maps of different resolution? For example, the final 8x8 feature maps might require a smaller window size.**
>
>
> We thank the reviewer for this thoughtful comment. We would like to clarify it is unnecessary to optimize the block size of LFC estimation.
> To be concrete, DIVISION (B=8, Q=2) is compared with normal training in the following table. It is observed that DIVISION has almost the same model accuracy as the normal training, which indicates a unified block size for all layers is enough for the training.
> It is unnecessary to further search the block size because the potential improvement of model accuracy is very limited (less than $0.4\%$), and it will increase the complexity and reduce the efficiency to search the block size.
>
>
> | Dataset | Architecture | Normal | DIVISION | Degradation |
> | :--: | :--: | :--: | :--: | :--: |
> | CIFAR-10 | ResNet-18 | $94.89$ | $94.7$ | $0.19$
> | CIFAR-10 | ResNet-164 | $94.90$ | $94.50$ | $0.4$
> | CIFAR-100 | DenseNet-121 | $79.75$ | $79.47$ | $0.28$
> | CIFAR-100 | ResNet-164 | $77.30$ | $76.90$ | $0.4$
> | ImageNet | ResNet-50 |  $76.15$ | $75.86$ | $0.29$
> | ImageNet | DenseNet-161 | $77.65$ | $77.58$ | $0.07$

---

> ### Author Response · Authors · 2022-11-18
> **To Reviewer PUeD**
>
> We thank the reviewer for the instructive comments on our work. We spent quite amount of time and effort in collecting more experiment results, trying our best to address the reviewer's concerns.
> As the deadline of context revising (Nov. 18th) is coming, would you please let us know if there are more questions/concerns left? It will be our pleasure to receive your follow-up feedback.

---

> ### Author Response · Authors · 2022-11-29
> **To Reviewer PUeD: Are there more questions/concerns left?**
>
> We thank the reviewer for the efforts in the paper review. We sincerely value your comments, and provide more explanations and experiment results to address your concerns. It has been two weeks since we posted our response. Would you please let us know if there are more questions/concerns left? It will be our pleasure to receive your follow-up feedback.

---

> > ### Comment · Reviewer_PUeD · 2022-12-07
> > **Thanks for the detailed feedback**
> >
> > Thanks for the feedback. Now I am somewhat more convinced on that the proposed method can also work for transformers / MLPs. Thus I raised my score.

---

> > > ### Author Response · Authors · 2022-12-07
> > > **Thank You**
> > >
> > > We sincerely thank the reviewer for spending time in the review and response.
> > > We are pleased that the concerns have been addressed, and we are encouraged that the reviewer gives a positive evaluation of our work.

---

### Official Review · Reviewer_xXpe · 2022-10-24

**Confidence:** 5
**Correctness:** 4
**Technical Novelty And Significance:** 3
**Empirical Novelty And Significance:** 3
**Recommendation:** 8

**Clarity, Quality, Novelty And Reproducibility:**

The code of the proposed method is available, what helps with reproducibility of the work. The work is clear and well explained. Overall the paper is of a good quality and presents a novel approach to the activation compressed training.

**Strength And Weaknesses:**

The proposed method is novel and interesting. The work is well structured. The problem is clearly introduced along with other existing methods, comparing their limitations and proposing ideas for improvements.  The stated hypothesis is analyzed from the experimental and theoretical perspectives.

The main weakness of the work is a limited number of datasets and topologies the method was evaluated with. It would be interesting to show that division into low and high frequency components can be applied to other problems as well. Specifically, it would be important to evaluate other types of layers that operate at the depth dimension, i.e., 1x1 convolutions.

Another limitation is the need selection/finding of other hyper parameters, i.e., B and Q. Some additional experiments showing that either they can be reused for other applications or quickly selected would be beneficial. Otherwise, the gain in training efficiency might not be that significant, if additional trainings have to be performed to find the best configurations.

Some other ideas for improvement of the method include the use of bfloat16 instead of float16 for storing LFC and the use of per tensor symmetric quantization.

**Summary Of The Paper:**

The presented work focuses on improving the efficiency of the training process by reducing memory cached during the backward propagation. Activation maps are split into low and high frequency components based on the observation that they don't affect the accuracy of the model equally. High precision copy of low frequency components is preserved, while the high frequency components are compressed, achieving over 10x compression of activation maps.


**Summary Of The Review:**

The work is interesting and presented claims are clear and supported by performed experiments. Some additional experiments would help to support it even more and show that it can be used in a wide range of applications, instead only for a small subset of solutions. Overall, the work is good in my opinion.

---

> ### Author Response · Authors · 2022-11-13
> **Response to Reviewer xXpe [Part 1/2 Q1]**
>
>
> We thank the reviewer for the thoughtful suggestions and detailed reviews.
>
> **[Q1.1] The main weakness of the work is a limited number of datasets and topologies the method was evaluated with. It would be interesting to show that division into low and high frequency components can be applied to other problems as well.**
>
> **[Q1.2] Specifically, it would be important to evaluate other types of layers that operate at the depth dimension, i.e., 1x1 convolutions.**
>
>
> **[Answer to Q1.1]**
>
> We thank the reviewer for this constructive comment. We conduct two experiments to show the performance of DIVISION on other types of datasets (Tabular dataset) and model topologies (MLP and Vision Transformer), respectively.
>
> The first experiment is conducted on the GAS dataset [1] (a tabular dataset, with 128-dimensional features, 13910 instances, and 6-class labels). The classification model is a 4-layer MLP (128 neurons in the input layer, 6 neurons in the output layer, and 64 neurons in the hidden layer). The model accuracy and memory cost (batch-size 256) are given in the following table. It is observed DIVISION achieves almost the same accuracy compared with normal training, with a $7.3\times$ compression rate. This indicates the effectiveness of DIVISION on the MLP models.
>
> | Method | Accuracy (\%) | Memory (KB) | Compression Rate
> | :-----: | :-----: | :-----: | :-----: |
> | Normal Training | $98.92$  |  $250$ | N/A |
> | DIVISION | $98.85$ |  $34.2$ | $7.3\times$ |
>
>
> The second experiment is conducted on the ImageNet dataset, where we adopt Swin Transformer-T [1] for the classification. The model accuracy, memory cost, and compression rate are given in the following table. It is observed that DIVISION achieves has $2.8\times$ compression rate, with nearly the same model accuracy compared with the normal training. This indicates DIVISION can effectively compress the training of the vision Transformer.
>
> | Method | Accuracy (\%) | Memory (GB) | Compression Rate
> | :----: | :----: | :----: | :-----: |
> | Normal Training | $81.2$ | $11.81$ | N/A |
> | DIVISION | $81.0$ | $4.19$ | $2.8\times$ |
>
> In conclusion, the evaluation based on multiple data types (tabular, and image) and model topologies (MLP, CNN, and Vision Transformer) indicate DIVISION can be widely applied to different scenarios.
>
>
> **[Answer to Q1.2]**
> To evaluate DIVISION with the depthwise convolution and pointwise convolution layers, we conduct experiments of MobileNet-V2 [2] on the CIFAR-10 and CIFAR-100 datasets, where the model accuracy is given in the following table. It is observed that DIVISION achieves nearly the same accuracy compared with normal training. This indicates the effectiveness of DIVISION for the depthwise convolution and pointwise convolution layers.
>
> | Method | MobileNet-V2 (CIFAR-10) | MoblieNet-V2 (CIFAR-100) |
> | :----: | :----: | :----: |
> | Normal Training | $91.9$ | $71.0$ |
> | DIVISION | $91.8$ | $70.6$ |
>
> [1] Ze Liu et al. Swin Transformer: Hierarchical Vision Transformer using Shifted Windows
>
> [2] Mark Sandler et al. MobileNetV2: Inverted Residuals and Linear Bottlenecks

---

> ### Author Response · Authors · 2022-11-13
> **Response to Reviewer xXpe [Part 2/2 Q2 and Q3]**
>
>
> **[Q2] Another limitation is the need selection/finding of other hyper parameters, i.e., B and Q. Some additional experiments showing that either they can be reused for other applications or quickly selected would be beneficial. Otherwise, the gain in training efficiency might not be that significant, if additional trainings have to be performed to find the best configurations.**
>
>
> We agree with the reviewer that it would be beneficial if we can transfer the hyper-parameter setting between different model architectures and datasets.
> To answer this question, we conduct experiments to use hyper-parameters $B\in\{8, 18\}$ and $Q\in\{2, 8\}$ to train ResNet-18 and MobileNet-V2 on CIFAR-10 and CIFAR-100. The model accuracy of different settings is given in the following table.
> It is observed $B$ and $Q$ have a consistent impact on different model architectures and datasets: the accuracy slightly grows with $Q$ and considerably reduces with $B$.
> This indicates we can reuse the hyper-parameter setting of DIVISION on CIFAR-10 to CIFAR-100 with the same model architecture, or we can reuse the setting with ResNet-18 and MobileNet-V2 on the same dataset. Moreover, according to our empirical study, $B=8$ and $Q=2$ can be a default setting which is effective for most model architectures and datasets.
>
> | | B=18 Q=2 | B=8 Q=2 | B=8 Q=8 |
> | :----: | :----: | :----: | :----: |
> | ResNet-18 CIFAR-10 | $78.7$ | $94.6$ | $94.9$ |
> | ResNet-18 CIFAR-100 | $73.2$ | $76.9$ | $77.0$ |
> | MobileNet-V2 CIFAR-10 | $10.0$ | $91.7$ | $91.0$ |
> | MobileNet-V2 CIFAR-100 | $62.4$ | $70.6$ | $71.6$ |
>
>
> **[Q3] Some other ideas for improvement of the method include the use of bfloat16 instead of float16 for storing LFC and the use of per tensor symmetric quantization.**
>
> We thank the reviewer for this constructive comment. We do observe a little improvement when using bfloat16 for storing the LFC, $\delta_l$, and $\Delta_l$, as shown in the following table. Bfloat16 can improve the model accuracy without memory overhead. We have updated our paper and code accordingly.
>
> |  | ResNet-164 CIFAR-10 | DensetNet-121 CIFAR-100 |
> | :----: | :----: | :----: |
> | bfloat16 | $94.58$ | $79.64$ |
> | float16 | $94.5$ | $79.5$ |
>
> Regarding the quantization methods, we conduct another experiment to study the memory cost (GB) of per-group (group size=256), per-channel, and per-tensor quantization (ImageNet dataset, batch-size 256), as shown in the following table. It is observed that the reduction of memory cost of per-tensor quantization is negligible compared with per-channel quantization. Moreover, we believe that Per-tensor quantization and Symmetric quantization may cause considerable degradation of model accuracy. Because different activation maps in a mini-batch may fall into different ranges, and each range may not be symmetric with the zero-point [1]. Therefore, we believe per-channel quantization is more suitable for DIVISION considering both model accuracy and memory cost. Anyway, we still appreciate it very much that the reviewer can share this comment with us.
>
> |  | Per-group Quantization | Per-channel quantization | Per-tensor Quantization |
> | :----: | :----: | :----: | :----: |
> | Act. Mem (GB) | $2.16$ | $1.97$ | $1.95$ |
> | Total Mem (GB) | $2.6$ | $2.41$ | $2.39$ |
>
> [1] Sharan Narang et al. Mixed Precision Training

---

### Official Review · Reviewer_R65G · 2022-10-25

**Confidence:** 3
**Correctness:** 3
**Technical Novelty And Significance:** 3
**Empirical Novelty And Significance:** 3
**Recommendation:** 5

**Clarity, Quality, Novelty And Reproducibility:**


The paper is well structured and the main idea is clearly described.

**Strength And Weaknesses:**

- Strengths

1. The paper introduce the concept of frequency domain to compress activation maps.

2. The authors theoretically and empirically show that the LFC component of activation maps are much more critical to maintain the model performance than the HFC component of them.

3. DIVISION consistently compresses activation maps with about 10x compression rate, while maintaining comparable model performance to normal training.



- Weaknesses

1. It is wondered whether $\lambda_l^{L} > \lambda_l^{H}$ in Theorem 1 can be guaranteed even when a different architecture (e.g., DenseNet-121) is utilized or the proportion of $W$ to $N$ is small (e.g., $W/N = 0.1$)

2. Even if $\lambda_l^{L} > \lambda_l^{H}$ in Theorem 1 can be guaranteed for different architectures and the small proportion of  $W$ to $N$, it is doubtful if Theorem 1 is still valid when DIVISION is exploited instead of DCT. In other words, when applying the average pooling rather than the inverse DCT, does Theorem 1 still hold so that $\text{GEB}_l^{L} < \text{GEB}l^{H}$? If not, the theoretical analysis might seem to be ineffective

3. When quantizing the HFC component of activation maps, the stochastic rounding is employed. Then, to obtain $V_l^{H}$ in Eq. (9), it is required to draw size($H_{l-1}$) samples, which seems to impose a significant burden on training. In addition, as $V_l^{H}$ varies depending on sampling, the performance of DIVISION might be likely to be different. Is there any performance degradation when using rounding instead of stochastic rounding?

**Summary Of The Paper:**

The authors of the paper observe that the high-frequency component (HFC) of activation maps possesses the majority of memory cost for training but the model performance can be preserved even in the absence of the HFC component of activation maps. Based on the observation, the paper proposes a new activation compressed training method called Dual Activation Precision (DIVISION) that keep the low-frequency component (LFC) of activation maps in high-precision while quantizing the HFC of activation maps into low-bit.

**Summary Of The Review:**

It is questionable whether Section 3 has something to do with Section 4 as mentioned in Weakness 1 and 2. I would encourage the authors to revise the paper by addressing aforementioned weaknesses.

---

> ### Author Response · Authors · 2022-11-13
> **Response to Reviewer R65G [Part 1/3 Q1]**
>
> We thank the reviewer for the thoughtful suggestions and detailed reviews.
>
> **[Q1] It is wondered whether $λ_l^L > λ_l^H$ in Theorem 1 can be guaranteed even when a different architecture (e.g., DenseNet-121) is utilized or the proportion of W to N is small (e.g., $W/N=0.1$)**
>
> We thank the reviewer for this instructive comment. To prove $λ_l^L > λ_l^H$ can be guaranteed for DenseNet-121, we conduct an experiment on the CIFAR-10 dataset. The values of $λ_l^L$ and $λ_l^H$ during the training (epochs 20, 40, and 60) of DenseNet-121 are given in the following tables, where the low-pass mask satisfies $W/N=0.1$ and $W/N=0.2$ in the first and second table, respectively; and $λ_l^L$ and $λ_l^H$ are estimated based on the input activation maps of the four DenseBlocks. It is consistently observed that $λ_l^L > λ_l^H$ for $W/N=0.1$ and $W/N=0.2$ in different training epochs. This indicates our proposed Theorem 1 holds without loss of generality.
>
> | Epoch | $20$ | $40$ | $60$ | Average $λ_l^L/λ_l^H$ |
> | :---------------------------------------: | :-----------------: | :------------: | :----------------: | :----------------: |
> | DenseBlock-1 | $λ_l^L=298.281, λ_l^H=218.605$   | $λ_l^L=184.913, λ_l^H=138.069$ |  $λ_l^L=142.668, λ_l^H=104.755$ | $1.36$ |
> | DenseBlock-2 | $λ_l^L=3.245, λ_l^H=1.713$       | $λ_l^L=1.284, λ_l^H=0.689$     |  $λ_l^L=0.687, λ_l^H=0.372$ | $1.87$ |
> | DenseBlock-3 | $λ_l^L=0.387, λ_l^H=0.260$       | $λ_l^L=0.160, λ_l^H=0.086$     |  $λ_l^L=0.084, λ_l^H=0.048$ | $1.70$ |
> | DenseBlock-4 | $λ_l^L=0.062, λ_l^H=0.009$       | $λ_l^L=0.011, λ_l^H=0.001$     |  $λ_l^L=0.006, λ_l^H=0.001$ | $7.56$ |
> | Average $λ_l^L/λ_l^H$ | $2.95$ | $3.12$ | $3.30$ | $3.12$ |
>
>
>
> | Epoch | $20$ | $40$ | $60$ | Average $λ_l^L/λ_l^H$ |
> | :---------------------------------------: | :-----------------: | :--------------: | :----------------: | :----------------: |
> | DenseBlock-1 | $λ_l^L=362.672, λ_l^H=154.214$   | $λ_l^L=225.543, λ_l^H=97.439$  | $λ_l^L=173.595, λ_l^H=73.828$  | $2.34$ |
> | DenseBlock-2 | $λ_l^L=3.632, λ_l^H=1.326$       | $λ_l^L=1.440, λ_l^H=0.533$     | $λ_l^L=0.774, λ_l^H=0.285$  | $2.72$ |
> | DenseBlock-3 | $λ_l^L=0.445, λ_l^H=0.202$       | $λ_l^L=0.179, λ_l^H=0.067$     | $λ_l^L=0.095, λ_l^H=0.037$  | $2.49$ |
> | DenseBlock-4 | $λ_l^L=0.062, λ_l^H=0.009$       | $λ_l^L=0.011, λ_l^H=0.001$     | $λ_l^L=0.006, λ_l^H=0.001$  | $7.56$ |
> | Average $λ_l^L/λ_l^H$ | $3.58$ | $3.78$ | $3.97$ | $3.78$ |

---

> ### Author Response · Authors · 2022-11-13
> **Response to Review R65G [Part 2/3 Q2]**
>
>
> **[Q2] Even if $\lambda_l^L > \lambda_l^H$ in Theorem 1 can be guaranteed for different architectures and the small proportion of $W$ to $N$, it is doubtful if Theorem 1 is still valid when DIVISION is exploited instead of DCT. In other words, when applying the average pooling rather than the inverse DCT, does Theorem 1 still hold so that $\mathrm{GEB}_l^L < \mathrm{GEB}_l^H$? If not, the theoretical analysis might seem to be ineffective.**
>
> We quite understand the review's concern on the validity of Theorem 1 when the average pooling is exploited for DIVISION. To bridge this gap, we have Theorem 2 and Remark 1 to demonstrate the moving average operation (average pooling) is essentially a low-pass filter. Although it is not absolutely the same as the result of the DCT-based filter, we believe that **Theorem 1 can be a proof-of-concept to indicate the HFC can be discarded with less accuracy degradation** compared with LFC, and can justify the effectiveness of DIVISION.
>
> The effectiveness of Theorem 1 can be validated by the following experiment results on CIFAR-10.
> In this experiment, the average pooling and DCT-based filters are both adopted for estimating LFC, and the model accuracy of each training method is given in the following table.
> It is observed that HFC-ACT consistently suffers from more degradation than LFC-ACT whichever algorithm is adopted for estimating the LFC. This is also consistent with the theoretical analysis that $\mathrm{GEB}^L < \mathrm{GEB}^H$. This indicates **the difference between average pooling and DCT-based filter is negligible in the analysis of DIVISION**.
>
> |  | Average Pooling | DCT-based filter |
> | :---: | :---: | :---: |
> | LFC-ACT | $82.95$ | $81.18$ |
> | HFC-ACT | $28.15$ | $29.52$ |
>
> Consider the advantages and disadvantages of these two low-pass filters. DCT has strong theoretical properties, but also has high computational complexity; while the moving average is efficient to estimate but its contribution to a DNN is difficult to theoretically analyze. Therefore, it is natural to employ DCT in the theoretical analysis, and deploy the average pooling in the algorithm of DIVISION.

---

> ### Author Response · Authors · 2022-11-13
> **Response to Review R65G [Part 3/3 Q3]**
>
> **[Q3] When quantizing the HFC component of activation maps, the stochastic rounding is employed. Then, to obtain $V_l^H$ in Eq. (9), it is required to draw size $(H_l−1)$ samples, which seems to impose a significant burden on training.**
>
> We thank the reviewer for this thoughtful comment. We would like to clarify that stochastic rounding will not cause too much burden to the training.
> To study the computational overhead of stochastic rounding, we conduct experiments on the CIFAR-10 dataset using a single RTX 3090 GPU. The training time of one epoch using stochastic rounding and deterministic rounding is given in the following table. It is observed **the computational overhead of stochastic rounding is negligible** during the training (ResNet-50: 0.22s for one epoch on a single GPU, 5.5s for 100 epochs on 4 GPUs).
>
> | Training time (1 epoch) | Deterministic rounding | Stochastic rounding |
> | :---------------------------------------: | :-----------------: | :---------------------------------: |
> | ResNet-18 |  $2.14 \pm 0.16$ | $2.18 \pm 0.15$ |
> | DenseNet-121 | $12.33 \pm 0.60$ | $12.63 \pm 0.59$ |
> | ResNet-50 | $9.32 \pm 0.44$ | $9.54 \pm 0.43$ |
>
>
> Moreover, despite the negligible speed improvement, deterministic rounding suffers from considerably higher rounding errors than stochastic rounding, which has been well-studied in existing work [1].
> Stochastic rounding is an unbiased operation, while deterministic rounding (a.k.a. round-to-nearest) is biased.
> We note that NNs will have multiple layers and the bias will propagate layer-by-layer and thus causing worse performance.
> Thus, stochastic rounding is the standard quantization method for quantized training frameworks, e.g. BNN [2], ActNN [3], and GACT [4].
> In this work, we follow existing work to adopt stochastic rounding for quantization.
>
> [1] M. Croci et al. Effects of round-to-nearest and stochastic rounding in the
> numerical solution of the heat equation in low precision.
>
> [2] Itay Hubara et al. Binarized Neural Networks.
>
> [3] Jianfei Chen et al. ActNN: Reducing Training Memory Footprint via 2-Bit Activation Compressed Training.
>
> [4] Xiaoxuan Liu et al. GACT: Activation Compressed Training for Generic Network Architectures.

---

> > ### Comment · Reviewer_R65G · 2022-11-15
> > **Response to rebuttal**
> >
> > Thank you for addressing my concerns. The responses to Q1 and Q2 are clear to me. However, I still have a concern about Q3.
> >
> > Although the computational overhead of stochastic rounding is negligible for ResNet-18, DenseNet-121, and ResNet-50, I am afraid that the burden resulting from stochastic rounding would be large when conducting experiments with large architectures (ex. ViT) on the ImageNet dataset as the batch size gets larger.
> >
> > I would appreciate if there is also no difference between deterministic rounding and stochastic rounding for experiments with large architectures (ex. ViT) on the ImageNet dataset as the batch size varies.

---

> > > ### Author Response · Authors · 2022-11-16
> > > **Response to Reviewer R65G**
> > >
> > > We thank the reviewer for this thoughtful feedback. To study the overhead of stochastic rounding on the vision transformers, we conduct experiments of viT-base-patch16-224 and Swin-Transformer-base-patch4-window7-224 on the ImageNet dataset using a single RTX 3090 GPU. The training time of one epoch using stochastic rounding and deterministic rounding is given in the following table. It is observed the overhead is less than the standard deviation of deterministic rounding (less than $0.2\%$ of deter. round.), which means they will have almost the same training time in most cases.
> > > This is consistent with our previous result, which indicates **the overhead of stochastic rounding is negligible for general model architectures**.
> > > We sincerely hope this can address the reviewer's concern.
> > >
> > > | Training time (1 epoch) | Deterministic rounding (sec) | Stochastic rounding (sec) | Overhead (percentage of deter. round.) |
> > > | :---: | :---: | :---: | :---: |
> > > | viT-base-patch16-224 |  $8078 \pm 56$ | $8091 \pm 58$ | $13$ (0.16%) |
> > > | Swin-Transformer-base-patch4-window7-224 |  $15608 \pm 25$ | $15615 \pm 39$ | $7$ (0.045%) |

---

> > > ### Author Response · Authors · 2022-11-29
> > > **To Reviewer R65G: Is Q3 solved?**
> > >
> > > We thank the reviewer for the feedback to our response. According to your comment, we have added one more experiment to demonstrate the overhead of stochastic rounding is negligible. Would you please let us know whether your concern about Q3 has been addressed?

---

> > > ### Author Response · Authors · 2022-12-07
> > > **To Reviewer R65G: Are there more questions/concerns left?**
> > >
> > > We thank the reviewer for the efforts in the paper review. We sincerely value your comments, and provide more explanations and experiment results to address your concerns. As the deadline of the discussion stage is coming, Would you please let us know if there are more questions/concerns left? It will be our pleasure to receive your follow-up feedback.

---

> > > ### Author Response · Authors · 2022-12-09
> > > **Last message to Reviewer R65G: Are there more questions/concerns left?**
> > >
> > > We thank the reviewer for the efforts in the paper review, and we have provided detailed explanations and experiment results to address your concerns. As the discussion stage will be ended in three days, would you please let us know if there are more questions/concerns left? We will believe no concern is left if you keep no more feedback on our response until the end of the discussion.

---

### Author Response · Authors · 2022-11-13
**Overall Response and Summarization of the Revision**

We thank all the reviewers for your time and constructive reviews. We are encouraged that they found our work to be novel (xXpe, PUeD, Eerz), well structured (R65G, xXpe, PUeD, Eerz), and insightful (Eerz).
It is our pleasure to improve this work according to your reviews.

First of all, we would like to highlight one contribution of DIVISION, that it proposes a **simpler** and **more transparent** framework than existing work. We believe improving the simplicity and transparency of ML algorithms is important for both the community and follow-up research, besides focusing on model accuracy and compression rate results.

We have revised the paper to address the concerns of reviewers. The modified parts are highlighted in blue color. The revisions are summarized as follows:

- (Reviewer R65G) We add the experiment of  $\lambda_l^L$  and  $\lambda_l^H$ estimation using DenseNet-121 with  $𝑊/𝑁=0.1$  and  $𝑊/𝑁=0.2$  in Appendix D.

- (Reviewers xXpe, PUeD, and Eerz) We add the experiment of Swin-Transformer on the ImageNet dataset to show the performance of DIVISION on vision transformers in Appendix L. We opensource the code at https://anonymous.4open.science/r/division-swin-transformer-F83E/README.md.

- (Reviewer xXpe) We add the experiment of MobileNet-V2 on the CIFAR-10 and CIFAR-100 datasets to show the performance of DIVISION on depthwise and pointwise convolutional layers in Appendix M.

- (Reviewer xXpe) We add a section in Appendix N to study the re-utilization of hyper-parameter settings across different model architectures and datasests.

- (Reviewer PUeD) We give the details of DIVISION in processing 1D/3D activation maps in Appendix F.

- (Reviewer PUeD) We add the experiment of MLPs on the GAS dataset (a tabular dataset) to show the performance of DIVISION on MLP models in Appendix K. We opensource the code at https://anonymous.4open.science/r/division-MLP-9416/README.md.

- (Reviewer PUeD) We add the experiment of to compare DIVISION with the Checkpoint strategy of Megatron-LM in Appendix O.

- (Reviewer Eerz) We update the throughput of all methods in Figures 1(a), 6(a) and 6(b), and the discussions in Section 5.3.

- (Reviewer Eerz) We add the experiment of fixed quantization under different bit-width in Appendix P.

- (Reviewer Eerz) We add the experiment to study the effect of batch-size on the throughput in Appendix Q.

Please let us know if you have any concerns about this work. We are looking forward to answering your follow-up questions.

---

### Author Response · Authors · 2022-12-02
**To area chairs, senior area chairs, and program chairs [Part 2/2]**

# About Reviewers' Concerns

We give the redirection links of our response to some key concerns of the reviewers here.  We sincerely hope all of the concerns have been addressed.

**[C1] Reviewer R65G wonders the overhead of stochastic rounding on vision transformer on the ImageNet dataset.**

Experiment results indicates the overhead of stochastic rounding is negligible (< 0.17%) for vision transformer on the ImageNet dataset.
The detailed response is given [[here]](https://openreview.net/forum?id=6Pv8AMSylux&noteId=nS3xOiSijJ).
We sincerely hope this concern has been addressed.

**[C2] Reviewers PUeD and Eerz wonder the performance of DIVISION on vision transformer.**

Experiment results on the Swin Transformer indicate that DIVISION achieves loss-less training (degradation < 0.3%), and higher compression rate ($2.8\times$) than Mesa ($2.2\times$).
The detailed response is given [[here]](https://openreview.net/forum?id=6Pv8AMSylux&noteId=9Cyo_1P4LtH).
We sincerely hope our response can address this concern.


**[C3] Reviewer PUeD wonders the performance of DIVISION on pure MLP**

Experiment results on the 4-layer MLP demonstrate that DIVISION provides loss-less training (degradation < 0.08%) for pure MLP models.
The detailed response is given [[here]](https://openreview.net/forum?id=6Pv8AMSylux&noteId=WVFfD7q3LtT).
We sincerely hope this concern has been addressed.


**[C4] Reviewer xXpe wonders the performance of DIVISION on depthwise convolution and pointwise convolution layers.**

Experiment results of MobileNet-V2 indicate that DIVISION provides loss-less training (degradation < 0.2%) for the depthwise convolution and pointwise convolution layers.
The detailed response is given [[here]](https://openreview.net/forum?id=6Pv8AMSylux&noteId=KlUsZqo8jy).
We sincerely hope our response can address this concern.

**[C5] Reviewer PUeD requests to compare DIVISION with the checkpoint strategy of Megatron-LM**

Experiment results indicate DIVISION achieves a higher compression rate than the checkpoint strategy of Megatron-LM on both CNNs (DIVISION $7.9\times$ vs Megatron-LM $2.27\times$) and vision transformers (DIVISION $2.8\times$ vs Megatron-LM $2.3\times$).
The detailed response is given [[here]](https://openreview.net/forum?id=6Pv8AMSylux&noteId=RAgxbxWAS4_).
We sincerely hope this concern has been addressed.


**[C6] Reviewer Eerz requests more experiment results of throughput and accuracy of fixed bit-width quantization.**

We conduct more experiments of throughput and fixed bit-width quantization, and we have given more analysis to the experiment results, including that of the throughput growing with the mini-batch size and the accuracy reducing with the bit-width.
All of these are are given [[here]](https://openreview.net/forum?id=6Pv8AMSylux&noteId=SEfVSxsZBI).
Our result is consistent with existing work [1].
We sincerely hope our response can address this concern.

# Major revision of our paper

We improved our paper following the directions of the reviewers.
The major revision is given [[here]](https://openreview.net/forum?id=6Pv8AMSylux&noteId=sD16SKHf7q).
We sincerely hope our revision can address the concerns of reviewers.

[1] Jianfei Chen et al. ActNN: Reducing Training Memory Footprint via 2-Bit Activation Compressed Training.

[2] AC-GC: Lossy activation compression with guaranteed convergence.
Advances in Neural Information Processing Systems

[3] Xiaoxuan Liu et al. GACT: Activation Compressed Training for Generic Network Architectures.

[4] "What is Occam's Razor?". math.ucr.edu. Retrieved 1 June 2019.

---

### Author Response · Authors · 2022-12-02
**To area chairs, senior area chairs, and program chairs [Part 1/2]**

# To area chairs, senior area chairs, and program chairs

We thank all of your efforts in the paper review of ICLR 2023. Moreover, we sincerely appreciate the reviewers can freely spend their time in the paper review.
Therefore, we spent quite amount of time and effort in the paper and experiments, respectfully replied to their comments, and tried our best to address their concerns.

================== Updated on 12/06/2022 ==================

We are pleased that Reviewer xXpe maintains an evaluation score of 8 with high confidence.
Moreover, we are encouraged to see Reviewer PUeD has raised the evaluation score according to our feedback and paper revision.

================== Updated on 12/13/2022 ==================

We are pleased that Reviewer Eerz has provided his/her feedback on 12/12/2022.
Although we have different opinions regarding some aspects, we have achieved a consensus on the merits of this work, i.e., motivation, overall idea, and simplicity.
Regarding the concern of reviewer Eerz, we believe it will play a crucial role in the outcome of our work.
Thus, we'd like to share our opinions [[here]](https://openreview.net/forum?id=6Pv8AMSylux&noteId=4voQ1KCs0td) for discussion purposes.
We sincerely hope our response can address the concerns of the reviewer and area chairs.

As for Reviewer R65G, we believe our responses have addressed his/her concerns because there is no further question/concern raised from his/her side till the end of rebuttal.
Again, we thank all of the reviewers and chairs for their time and effort in the paper review.


# Our Research Motivation

We believe existing ACT algorithms are over-complicated.
e.g. searching the quantization bit-width during DNN training in existing work [1,2,3] is unnecessary, where a fixed bit-width is enough for loss-less training in our algorithm.
An over-complicated framework will not only increase the computational overhead (reduces the efficiency), but also make itself less transparent. This brings
challenges to the real-world applications. e.g. deploying a non-transparent algorithm will too much rely on the experiment results.

According to the theory of Occam's razor in ML, the entities should not be multiplied beyond necessity [4]. e.g. it is unnecessary to search the bit-width during the training if a fixed bit-width is enough.
Therefore, we believe improving the simplicity and transparency of ML algorithms is important for both the community and follow-up research; and we propose a simple (efficient), transparent (insightful), and effective (loss-less training) memory-efficient training method DIVISION in our paper.


# Our Contribution


First, we have an instructive observation in our preliminary experiments:
**DNN backward propagation mainly utilizes the LFC of the activation maps, while the majority of memory is for caching the HFC during the training.**
Beyond the experiment results, we also theoretically prove this phenomenon (DNN backward propagation mainly utilizes the LFC) widely exists in DNN training.

The preliminary result and theoretical analysis allow us to understand the compressible (HFC) and non-compressible factors (LFC) during DNN training.
Following this direction, we propose a simple framework DIVISION to effectively reduce the memory cost of DNN training via removing the redundancy in the HFC during the training.
DIVISION has very simple strategies to compress the HFC, e.g. fixed bit-width and per-channel quantization.
Such simple compression of HFC works without degradation of accuracy due to the fact that HFC has less contribution to DNN backward propagation (our preliminary result).

Therefore, we believe our work has a non-trivial contribution to the ML community and follow-up research, and we are encouraged that Reviewers xXpe and PUeD recognize our contribution with high confidence.

---

### Author Response · Authors · 2022-12-11
**Summarization of Rebuttal**

# Summarization of Rebuttal

We thank the reviewers and chairs for their efforts and time in the paper review.
We are pleased that the reviewers found our work to be novel (xXpe, PUeD, Eerz), well structured (R65G, xXpe, PUeD, Eerz), insightful (Eerz), and simple and effective (PUeD).
After the rebuttal, we are pleased that Reviewer xXpe maintains an evaluation score of 8 with high confidence; and Reviewer PUeD has raised the evaluation score to 6 according to our feedback and revision.

Although we have different opinions regarding some aspects with reviewer Eerz, we have achieved a consensus on the merits of this work, i.e., motivation, overall idea, and simplicity.
Regarding the concern of reviewer Eerz, we have shared our response [[here]](https://openreview.net/forum?id=6Pv8AMSylux&noteId=4voQ1KCs0td) for discussion purposes.
We sincerely hope our response can address the concerns of the reviewer and area chairs.

As for Reviewer R65G, we believe our responses have addressed his/her concerns because there is no further question/concern raised from his/her side till the end of rebuttal.
For the convenience of going through our responses, we give the redirection links to each of our response in the following table, including our research motivation and contribution, overall response and major revision, response to some key concerns, and detailed response to all reviewers.
We kindly request the chairs could carefully go through our responses, and we sincerely respect the final decision of the chairs.


| Content | Redirection Link |
| :--: | :--: |
| Our research motivation | [[here]](https://openreview.net/forum?id=6Pv8AMSylux&noteId=CAh1Qd17fRM)
| Our contribution | [[here]](https://openreview.net/forum?id=6Pv8AMSylux&noteId=CAh1Qd17fRM)
| Overall response and major revision | [[here]](https://openreview.net/forum?id=6Pv8AMSylux&noteId=sD16SKHf7q)
| Response to some key concerns | [[here]](https://openreview.net/forum?id=6Pv8AMSylux&noteId=PWdMmOXYGr)
| Detailed response to Reviewer R65G | [[here]](https://openreview.net/forum?id=6Pv8AMSylux&noteId=bYdQjOXgZl)
| Detailed response to Reviewer xXpe | [[here]](https://openreview.net/forum?id=6Pv8AMSylux&noteId=dc4esgwMpg)
| Detailed response to Reviewer PUeD | [[here]](https://openreview.net/forum?id=6Pv8AMSylux&noteId=02YiRAMID1u)
| Detailed response to Reviewer Eerz | [[here]](https://openreview.net/forum?id=6Pv8AMSylux&noteId=TLD2SJhyObr)

---

### Author Response · Authors · 2022-12-13
**Empirical Comparison with GACT**

According to the official paper of [[GACT]](https://proceedings.mlr.press/v162/liu22v/liu22v.pdf), GACT has a compression rate of $11.39 \times$ on ResNet-50, which is slightly higher than DIVISION ($10.4\times$ on ResNet-50).
However, DIVISION has higher training throughput than GACT.
Specifically, GACT has nearly the same training throughput as ActNN according to Fig. 4 (a) in the paper of GACT.
DIVISION has higher throughput than ActNN according to [[here]](https://openreview.net/forum?id=6Pv8AMSylux&noteId=WJgaW0c8Ar).
Therefore, by adopting ActNN as a common medium, it is demonstrated that DIVISION has higher training throughput than GACT.
The high training throughput of DIVISION derives from its simplicity.

---

### Author Response · Authors · 2022-12-13
**A follow-up response to the feedback of reviewer Eerz**

Dear AC and reviewers:

We sincerely appreciate the reviewers and chairs for spending their time on the paper reviews. Having a current score of `8653` with reviewer R65G (score: 5) being pleased with our rebuttal (pending a straightforward concern we already resolved with new experiment results [[here]](https://openreview.net/forum?id=6Pv8AMSylux&noteId=nS3xOiSijJ)). We believe the concerns of reviewer Eerz will play a crucial role in the outcome of our work — thus, we'd like to share our opinions here for discussion purposes.


Although we and reviewer Eerz have different opinions regarding some aspects, we have achieved a consensus on the merits of this work, i.e., motivation, overall idea, and simplicity. However, regarding the empirical evaluation, we respectfully disagree with the following claims of reviewer Eerz.


- **The claim that DIVISION may only show advantages in complicated architectures is factually untrue, verifiable with empirical evidence.**
According to our paper and response, DIVISION is evaluated on general vision model architectures consist of different complexities, including [[ResNet-50]](https://openreview.net/forum?id=6Pv8AMSylux&noteId=15HS27fmQf), WRN-50-2 (in the paper), [[MobileNet]](https://openreview.net/forum?id=6Pv8AMSylux&noteId=dSwsnhgc7D), [[Swin-transformer]](https://openreview.net/forum?id=6Pv8AMSylux&noteId=RAgxbxWAS4_), and [[MLP]](https://openreview.net/forum?id=6Pv8AMSylux&noteId=WVFfD7q3LtT). Both
Reviewers xXpe and PUeD have appreciated the effectiveness of DIVISION on general vision models.


- **The claim that our ActNN results were incorrectly reproduced is ungrounded, again backed by reported results.**
We would like to remark that our reproduced throughput of ActNN is consistent with the reviewer-provided result (266 image/s (our) vs 280 images/s (reviewer), see [[here]](https://openreview.net/forum?id=6Pv8AMSylux&noteId=WJgaW0c8Ar)).
We believe our experiment settings and results are well-founded, which have been given in Appendix I and [[github]](https://anonymous.4open.science/r/division-5CC0/).
Both Reviewers xXpe and PUeD endorsed the good reproducibility of our work.


- **ACT's simplicity comes at the cost of accuracy loss; DIVISION is simple yet lossless.** It has been proved in existing work BLPA and ACTNN that a basic ACT framework suffers from accuracy loss.
To have a lossless ACT, ActNN, AC-GC, and Mesa focus on searching the quantization parameters during training, which is complicated and non-transparent.
DIVISION is a simpler and more transparent framework than these pieces of SOTA work, without accuracy loss.


- **Existed experiments already showcased the SOTA nature of DIVISION, passing standard top conferences' scrutiny.**
Specifically, DIVISION has been compared with **6** SOTA methods spanning from the year 2019 to 2022, including BLPA [1], AC-GC [2], ActNN [3], Checkpoint of Megatron-LM [4], SWAP [3], and Mesa [5].
According to some recent outstanding work published in top conferences [5, 6, 7] and a quick survey of ACT papers submitted to ICLR23 [8, 9, 10], all abovementioned methods are compared with **$\leq$ 4 SOTA methods without GACT.**
Therefore, while we believe GACT can be a nice addition as more experiments are always better, the lack of it cannot be a fairground for rejection, nor a score of 3, given the rich comparison we have already done.

  The lack of GACT comparison is mostly due to time/resource limitations: we already carried out a hefty rebuttal with **eight** experiments added, addressing more crucial concerns. For the spirit of discussion, [[here]](https://openreview.net/forum?id=6Pv8AMSylux&noteId=vjcY0SaOBV) we offer an empirical comparison with GACT via a common medium. Should the decision comes down to this particular experiment, we are willing to replicate then add it to our camera-ready version.


To this end, we sincerely hope the chairs and reviewers may consider the contribution of this work based on our responses, and we respect the committee's final decision.


Yours sincerely,

Authors


[1] Backprop with approximate activations for memory-efficient network training. NeuraIPS 2019

[2] AC-GC: Lossy activation compression with guaranteed convergence. NeuraIPS 2021

[3] ActNN: Reducing Training Memory Footprint via 2-Bit Activation Compressed Training. ICML 2021

[4] Megatron-LM: Training multi-billion parameter language models using model parallelism." arXiv:1909.08053. 2019

[5] Mesa: A Memory-saving Training Framework for Transformers." arXiv:2111.11124. 2021

[6] Back Razor: Memory-Efficient Transfer Learning by ...... NeuraIPS 2022

[7] Block-Wise Dynamic-Precision Neural Network Training Acceleration via ...... ASP-DAC23

[8] FEW-BIT BACKWARD: QUANTIZED GRADIENTS OF ACTIVATION FUNCTIONS FOR MEMORY FOOTPRINT REDUCTION

[9] LEARNING WITH AUXILIARY ACTIVATION FOR MEMORY-EFFICIENT TRAINING

[10] WINNING BOTH THE ACCURACY OF FLOATING POINT ACTIVATION AND THE SIMPLICITY OF INTEGER ARITHMETIC

---

### Decision · Program_Chairs · 2023-01-20

**Decision:**

Reject

**Justification For Why Not Higher Score:**

I need to discuss further with the SAC.

----Updates----

After discussing with SAC, we decided to recommend a rejection for this paper. Below are comments provided by SAC.

- The reviewer Eerz agreed that the idea is interesting but experimental evaluation seems to have some critical issues.
- Evaluation parts are so important, so if there is some critical issue here, it should be a good reason to reject.
- He is an assistant professor in a prestigious university, so I believe he is a good active researcher and reviewer. Indeed, his review quality is great too.
- The other reviews are not so enthusiastic for this work, given that one reviewer has a 'marginal reject' opinion and another has 'marginal accept'.


**Justification For Why Not Lower Score:**

N/A

**Metareview: Summary, Strengths And Weaknesses:**

This paper is based on a nice observation that the high-frequency component (HFC) of activation maps possesses the majority of memory cost for training but the model performance can be preserved even in the absence of the HFC component of activation maps. Then the authors propose a new activation compressed training method called Dual Activation Precision (DIVISION) that keep the low-frequency component (LFC) of activation maps in high-precision while quantizing the HFC of activation maps into low-bit.

Strengths:
1. The proposed method is novel and interesting. The paper is well structured.
2. The evaluation is quite comprehensive. After the rebuttal, a lot of experimental results are included, e.g., DIVISION on ViT, depthwise convolution and pointwise convolution layers.

Weaknesses:
1. Compared with a recent baseline called GCAT, DIVISION achieves lower compression rate.
2. Reviewer Eerz has a strong concern on the results reported. He ran the codes and found some results are not correct in the paper. For this reason, he is skeptical about other results in the paper.


**Summary Of Ac-Reviewer Meeting:**

We conducted online meetings with Reviewers Eerz, xXpe and PUeD on 13 Dec. R65G did not join the meeting, but he emailed me that he is good with the responses and the authors had addressed the concerns from different reviewers.

PUeD is good with the responses from the authors, and thus he has already increased his score from 5 to 6. Meanwhile he has some new concerns about the average pooling, but he will not change the score.

Eerz has concerns about the experiments. After he found incorrect results reported in the paper, he was quite skeptical about the results. He agreed with other reviewers about the novelty, as well as the great efforts to revise the paper during rebuttal. However, his suggestion is to reject the paper, so that the authors have more time to carefully check all the codes and re-generate the results.

xXpe shared her thoughts about Eerz's concern. First, she likes the efforts of the authors during the rebuttal – they had conducted quite a number of new experiments to address the concerns from reviewers.  Second, we should not reject a paper because of some suspects without evidence.

I concluded that I actually agree with both Eerz and xXpe. I need to discuss with Senior AC about this paper.